# Role of Hydroxytyrosol and Oleuropein in the Prevention of Aging and Related Disorders: Focus on Neurodegeneration, Skeletal Muscle Dysfunction and Gut Microbiota

**DOI:** 10.3390/nu15071767

**Published:** 2023-04-04

**Authors:** Laura Micheli, Laura Bertini, Agnese Bonato, Noemi Villanova, Carla Caruso, Maurizia Caruso, Roberta Bernini, Felice Tirone

**Affiliations:** 1Institute of Biochemistry and Cell Biology, National Research Council (IBBC-CNR), Via E. Ramarini 32, Monterotondo, 00015 Rome, Italy; 2Department of Ecological and Biological Sciences (DEB), University of Tuscia, Largo dell’Università, 01100 Viterbo, Italy; 3Department of Agriculture and Forest Sciences (DAFNE), University of Tuscia, Via San Camillo de Lellis, 01100 Viterbo, Italy

**Keywords:** hydroxytyrosol and oleuropein chemistry, pharmacokinetics, brain neurodegeneration, neuroprotection, aging, gut–brain axis and microbiota, metabolism, muscle dysfunctions, senescence

## Abstract

Aging is a multi-faceted process caused by the accumulation of cellular damage over time, associated with a gradual reduction of physiological activities in cells and organs. This degeneration results in a reduced ability to adapt to homeostasis perturbations and an increased incidence of illnesses such as cognitive decline, neurodegenerative and cardiovascular diseases, cancer, diabetes, and skeletal muscle pathologies. Key features of aging include a chronic low-grade inflammation state and a decrease of the autophagic process. The Mediterranean diet has been associated with longevity and ability to counteract the onset of age-related disorders. Extra virgin olive oil, a fundamental component of this diet, contains bioactive polyphenolic compounds as hydroxytyrosol (HTyr) and oleuropein (OLE), known for their antioxidant, anti-inflammatory, and neuroprotective properties. This review is focused on brain, skeletal muscle, and gut microbiota, as these systems are known to interact at several levels. After the description of the chemistry and pharmacokinetics of HTyr and OLE, we summarize studies reporting their effects in in vivo and in vitro models of neurodegenerative diseases of the central/peripheral nervous system, adult neurogenesis and depression, senescence and lifespan, and age-related skeletal muscle disorders, as well as their impact on the composition of the gut microbiota.

## 1. Introduction

Aging is a complex process, caused by the time-dependent accumulation of cellular damage, that is linked to a progressive decline in physiological functions at the tissue and organismal level. This deterioration leads to an impaired capacity to respond to challenges and a higher risk to contract major human pathologies, such as cognitive decline, neurodegenerative and cardiovascular diseases, cancer, diabetes, sarcopenia, and osteoporosis.

With age, the chronic response to lifelong stress leads to a physiological inflammation state, conserved in evolution and important for survival. Such a systemic low-grade inflammation state, termed “inflammaging”, is highly relevant to the pathogenesis of the majority of age-associated degenerative diseases [1]. It has been suggested that in successfully aging individuals, there is a balance between pro-inflammatory stimuli (inflammaging)—such as interleukins IL-15, IL-18, IL-6, serum-amyloid A, fibrinogen, C reactive protein (CRP), von Willebrand factor, resistin, and leukotrienes [2,3,4]—and anti-inflammatory stimuli, such as adiponectin, the arachidonic acid cascade (cytochrome p450), IL-1 receptor antagonist (IL-1RA), and soluble tumor necrosis factor receptor (STNF-R) [5,6,7].

The increasing understanding of the molecular basis of aging allowed to define twelve candidate hallmarks that are common denominators of aging in different organisms. These hallmarks are genomic instability, telomere attrition, epigenetic alterations, loss of proteostasis, disabled macroautophagy, deregulated nutrient sensing, mitochondrial dysfunction, cellular senescence, stem cell exhaustion, altered intercellular communication, chronic inflammation, and dysbiosis [8].

Several features of aging appear to involve autophagy, a conserved catabolic mechanism that maintains cellular homeostasis by targeting for degradation and recycling damaged organelles, toxic protein aggregates, and other dysfunctional cellular components [9]. The autophagic capacity declines with age as the result of deregulation of upstream nutrient-sensing signaling pathways that have been shown to modulate the rate of aging in various vertebrate and invertebrate species [10]. In particular, aging is associated with aberrant activation of mechanistic target of rapamycin (mTOR), an anabolic, pro-aging pathway that suppresses autophagy [11]. Furthermore, dysregulation of the insulin/insulin growth factor 1 (IGF1) signaling pathway during aging contributes to suppressing autophagy via activation of mTOR signaling and inhibition of forkhead box O (FOXO) transcription factors that are known to promote the autophagic process through several mechanisms [12].

Conversely, the AMP activated kinase (AMPK) and the silent mating type information regulation 2 homolog 1 (Sirtuin1, SIRT1) catabolic pathways, which signal nutrient scarcity and stimulate autophagy, are downregulated with age. AMPK and SIRT1 have a close interaction in the regulation of energy metabolism and aging since they can reciprocally enhance each other’s activity. They have been shown to extend the lifespan of numerous organisms and have other anti-aging effects by exerting multiple actions [13,14]. Briefly, these actions include: (1) repression of inflammation through inhibition of nuclear factor-kB (NF-kB) pro-inflammatory signaling, whose continued activation promotes the aging process [15]; (2) stimulation of autophagy through direct targeting of components of the autophagy machinery, repression of mTOR signaling and activation of FOXO factors; (3) maintenance of mitochondrial homeostasis and downregulation of reactive oxygen species (ROS) generation through activation of peroxisome proliferator-activated receptor coactivator 1α (PGC1α), a principal transcriptional coactivator involved in the regulation of mitochondrial biogenesis and function; (4) defense against oxidative stress through activation of FOXO and mammalian nuclear factor erythroid-derived 2 (NEF2l2, also known as Nrf2) transcription factors. Furthermore, SIRT1 can suppress cellular senescence by preventing telomere shortening, by promoting DNA damage repair, and by stabilizing the chromatin structure through the deacetylation of a variety of histone and non-histone proteins [16].

Based on this knowledge, new pharmacological approaches to counteract aging and prevent chronic degenerative diseases focus on drugs or natural compounds capable of activating the anti-aging AMPK and SIRT1 pathways and inhibiting the mTOR and insulin/IGF1 pro-aging pathways [17].

Beyond pharmacological approaches, diet and lifestyle have emerged as important factors to counteract and restrain the degenerative processes related to aging. A large body of evidence has shown close relationships between the Mediterranean Diet (MD) and longer survival and prevention of age-related diseases [18]. Extra virgin olive oil (EVOO), an important component of MD, is rich in bioactive compounds such as polyphenols, which are well-known to possess antioxidant, anti-inflammatory, and neuroprotective properties [19,20] (Hu et al., 2014; Visioli et al. 2020).

In this review, we summarize studies reporting the effects exerted by two major EVOO polyphenols, hydroxytyrosol (HTyr) and oleuropein (OLE), on age-related neurodegenerative diseases, skeletal muscle-wasting and associated bone disorders, and alterations in the intestinal microbiota composition in healthy or age-related pathological states.

We focus on brain, skeletal muscle, and gut microbiota given the existence of multiple interactions between these systems. In fact, in addition to direct nerve–muscle interactions through the neuromuscular junction, skeletal muscle secretes growth factors and cytokines (commonly termed myokines) that mediate direct crosstalk between muscle and brain functions [21].

Furthermore, there is ample evidence of bidirectional communications between brain and gut, involving the microbiota, that comprise neuronal (e.g., enteric nervous system, vagus, and sympathetic and spinal nerves) and humoral pathways (e.g., cytokines, hormones, and neuropeptides as signaling molecules). Such a communication network is referred to as the “gut microbiota–brain axis” and may contribute to behavior and neurodegenerative disorders [22]. Likewise, a number of recent studies in animal models and in humans support the existence of a gut–muscle axis that may regulate the onset and progression of age-related physical frailty and sarcopenia. Alterations in the gut microbiota composition could in fact promote chronic inflammation and anabolic resistance, ultimately causing reduced muscle mass, impaired muscle function and adverse clinical outcomes [23,24,25].

This review begins with a survey on HTyr and OLE chemistry, pharmacokinetics, and metabolism, to elucidate their pathways of synthesis.

## 2. Chemistry of Hydroxytyrosol (HTyr) and Oleuropein (OLE)

In the last few years, scientific research has highlighted how numerous epidemiological and dietary studies ascribe to plant foods rich in bioactive chemical compounds a direct association with the reduced onset of chronic and neurodegenerative diseases [26]. As demonstrated by a wide number of studies, fruits and vegetables prevent the occurrence of several types of cancer, inflammation, cardiovascular, and neurodegenerative diseases [27,28,29,30]. They also counteract the aging processes [31].

Many of these healthy effects are associated to the daily intake of EVOO, a pillar food of the MD [32,33]. Numerous studies demonstrated that EVOO plays a cardioprotective role, exerts a protective effect against atherosclerosis, lowers blood pressure, improves cognitive performance, verbal fluency, and visual memory, glucose metabolism, and reduces the occurrence of Parkinson’s and Alzheimer’s diseases [19,20].

The beneficial effects of EVOO are associated with its peculiar chemical composition [34], which includes a saponifiable fraction (98–99%) and an unsaponifiable (non-polar) and polar fraction (1–2%). All compounds found in both fractions are responsible for the beneficial properties of EVOO and make it a functional food.

The saponifiable fraction is composed by triacylglycerols containing mainly monounsaturated fatty acids. The major is oleic acid (55–83%); the minor is palmitoleic acid (0.3–3.5%) as reported in Figure 1. These compounds are responsible for many beneficial effects on human health [35].

The unsaponifiable (non-polar) and polar fraction is very heterogeneous including a variety of structurally distinct compounds as terpenes, sterols, and non-polar and polar phenols. Among them, phenols play a central role for human health, countering non-communicable diseases [36]. Non-polar phenols include tocopherols, commonly known as vitamin E, which contribute to protect cells from oxidative processes [37].

Polar phenols, commonly known as polyphenols, include phenolic acids (benzoic and cinnamic acids), phenolic alcohols and their derivatives, secoiridoids and their aglycons, lignans, flavonoids, and hydroxychromones.

Organoleptic characteristics and health properties of EVOO depend on the qualitative and quantitative phenolic profile. Varying olives cultivar, climate, stage of maturity, time of harvesting, and EVOO production process, the total polyphenol concentration significantly changes from 20 to 600 mg/kg [38].

Among these compounds, HTyr and OLE are extensively studied for their biological and pharmacological properties [39] (Figure 2).

OLE is a secoiridoid found in many plants. In the *Oleacea* family, its content is high in leaves and green olives (26.5 and 19.2 mg/Kg, respectively). During fruit ripening, the content of OLE decreases being hydrolyzed by endogenous α-glycosidases producing glucose, oleuropein aglycone, HTyr and elenolic acid (Figure 1). A similar process occurs during EVOO production [40].

In olive plants, OLE contributes to the quality of the olives and exhibits antibacterial activity, conferring resistance to many pathogens [41].

Due to high bioavailability and a high degree of absorption, OLE is a promising nutraceutical with antioxidant, antidiabetic, cardioprotective, antiatherogenic, and neuroprotective activity [42,43].

HTyr is one of the main phenolics found in EVOO; its content varies from 3.0 to 25.6 mg/Kg. This small phenol is a powerful antioxidant, and this effect is related to the presence of the catecholic moiety which was converted into the corresponding quinone [44].

HTyr has a high potential therapeutic effect exhibiting anticancer, cardioprotective, antidiabetic, anti-inflammatory, and neuroprotective activities [45,46,47,48].

Being a hydrophilic compound, HTyr can be recovered by olive oil wastes and by-products. Sustainable procedures such as membrane technologies are afforded to HTyr-enriched extracts [49,50]. These processes are according to the circular economy strategy because wastes and by-products become a precious source of high-added-value compounds [51].

To carry out in vivo studies, pure HTyr was prepared by different synthetic methods starting from commercially available compounds such as 3,4-dihydroxyphenylacetic acid [52] or tyrosol using chemical or enzymatic procedures [53,54,55,56].

Synthetic procedures have been also optimized to improve the bioavailability of HTyr and OLE. Lipophilic derivatives [57,58,59,60,61] and novel compounds were synthetized without modifying the catecholic moiety to be tested in vitro and in vivo as antioxidant, anticancer, anti-inflammatory, and neuroprotective agents [62,63,64,65]. Emerging research is devoted to developing drug delivery systems based on liposomes [66] and nanotechnology [67] for a better bioavailability and improved biological activities of both HTyr and OLE [68].

## 3. Pharmacokinetics of HTyr and OLE

### 3.1. Absorption

After ingestion of EVOO, a micellar solution composed by an aqueous and a lipid phase is produced. A first modification takes place in the mouth due to the hydrolytic action of the saliva; therefore, the polyphenols reach the stomach where they are partially hydrolyzed before passing to the small intestine. While OLE reaches the small intestine together with HTyr, the corresponding Ole aglycone is susceptible to the environment and residence time of the stomach (pH = 2.0, up to 4 h), where it is hydrolyzed maximizing the rise of free HTyr and elenolic acid (Figure 1) [69,70].

Being a glycoside, OLE sometimes undergoes further absorption pathways; in addition to passive and paracellular diffusion, it can also be taken up into epithelial cells of the small intestine via a sodium-dependent glucose transporter 1 (SGLT1) localized to the apical side of cells, used as an active transporter to move glucose to the basal zone. Thus, OLE accesses the SGLT1 transporter, crosses the epithelial cell, and then reaches the bloodstream via the Glut2 transporter by facilitated diffusion (Figure 3) [71,72].

HTyr is the best absorbed phenolic compound in the intestinal tract with an efficiency ranging from 40% to 95% depending on the residence time, reaching the peak concentration in human plasma 1 h after ingestion [73]. HTyr uptake follows passive bidirectional diffusion across the human enterocyte membrane [72]; moreover, the food matrix influences its absorption: oily matrices allow the absorption of HTyr more easily than aqueous solutions or yogurt [74,75]. Phenolic bioavailability is known to be influenced by several factors such as age, hormonal status, or gender, as described in the literature [34,76].

### 3.2. Metabolism and Distribution

Despite the absorption processes of OLE described above, the potential therapeutic efficacy of this molecule is interesting. Several preclinical studies have shown that OLE is a ligand of the peroxisome proliferator-activated receptor α (PPAR α) that regulates cytochrome P450 and controls lipid homeostasis, the activation of which mediates the pleiotropic effects of various vital functions of the body, making OLE a cardioprotective and neuroprotective agent [77]. In particular, OLE can act both as a substrate of P450s and as their inducer or inhibitor [78]. These properties lead to the consideration that OLE, in combination with drugs that are substrates of P450s, may modify their pharmacokinetic and therapeutic efficacy.

Once absorbed, HTyr is distributed through capillaries mainly to the muscles, testicles, liver, kidneys, and brain, with renal uptake 10 times higher than other organs [39,79], reaching the maximum plasma concentration about 7 min after oral intake [80].

Recent research suggests that the highest plasma concentrations are reached between 30 min and 1 h after oral administration, being no longer detectable after 4 h [81]. Once absorbed, HTyr binds to plasma high-density lipoprotein (HDL), carrying out its antioxidant and cardioprotective capacity [82]. Its metabolism is rapid and has a plasma half-life of about 1–2 min; despite this, it shows excellent distribution capabilities in various tissues, accumulating mainly in the kidneys and liver [83,84]. Its widespread distribution makes HTyr responsible for the many beneficial properties that distinguish it. Some authors demonstrated that after oral administration on rats of increasing doses of HTyr (1, 10, and 100 mg/kg), the main accumulation was in the liver, kidneys, and brain as well as in urine and plasma [85].

Enzymes involved in Phase II metabolism include sulfotransferase (SULF), acetyltransferase (ACT), uridine 5′-diphosphoglucuronosyl transferase (UGT), catechol-O-methyltransferase (COMT), which catalyze reactions giving rise to the corresponding sulfated, glucuronic, methylated, and acetylated HTyr derivatives, as depicted in Figure 2 [86,87,88].

HTyr appears to be the direct substrate of SULF and ACT enzymes, leading to the production of HTyr-4′-O-sulphate and HTyr acetate, respectively; in addition, SULF enzyme can produce HTyr acetate-4′-O-sulphate, homovanillyl alcohol-4′-O-sulphate and homovanillic acid-4′-O-sulphate [20,26,39]. UGT can act directly on HTyr, conjugating a glucuronide group in the 4′ position and on the derived metabolites, leading to the production of homovanillyl alcohol-4′-O-glucuronide and homovanillic acid-4′-O-glucuronide. HTyr sulphate, HTyr acetate-4′-O-sulphate, and HTyr-4′-O-glucuronide are the major metabolites detected in plasma and urine after daily consumption of HTyr [26,83,89]. Direct methylation of HTyr by COMT affords to homovanillyl alcohol (3-hydroxy-4-methoxyphenylethanol) that, by oxidation reaction, was converted into homovanillic acid (4-hydroxy-3-methoxyphenylacetic acid), resulting the reduction product of 3,4- dihydroxyphenylacetic acid (DOPAC), a dopamine metabolite [90].

Some in vivo studies have shown that the activation of SULF and UGT enzymes depends on the dose of administered HTyr. At low dose (1 mg/kg, 25–30%), the glucuronidation is the preferred via (86% glucuronidation; 14% of sulphation); on the contrary, at high doses (100 mg/kg), the sulphation is the main pathway (75% sulphation; 25% glucuronidation), demonstrating a dose-dependent activity exerted by HTyr [88,91].

The kidneys and liver exhibit the highest absorption of HTyr and its metabolites. HTyr acetate and HTyr acetate-4′-O-sulfate, due to their increased lipophilic character, cross the intestinal bilayer membrane more efficiently; consequently, in the plasma they have the highest concentration. Then, when HTyr is orally administered as acetate, it persists longer in plasma (0.5–2 h) [81].

### 3.3. Excretion

The excretion phase of HTyr and its metabolites follow predominantly through the kidneys depending on the concentration and method of administration [91,92]. The time required for the complete elimination from the body of both HTyr and its metabolites is about 6 h in humans and about 4 h in rats [81,91,93]. The compounds not absorbed by the intestine are excreted in the naturally ingested form or as products of chemical transformations in feces [94].

The excretion phase has been studied by various authors. Radiolabeling pure HTyr with ^3^H and administering it to male rats intravenously and orally in oily and aqueous formulations, some authors demonstrated that 94.1% of HTyr was recovered from urine samples from oily oral administration, 70.9% from aqueous oral administration and 94.9% from intravenous administration. This suggests that the orally administered oil-based formulation resulted in a higher elimination rate in the urine than the water-based formulation [76].

Other authors investigated the excreted metabolites of HTyr by feeding mice pure radiolabeled HTyr, demonstrating that the main forms found in the urine 5 h after administration were the sulphate-conjugated ones. Furthermore, they have highlighted the nephroprotective action of HTyr because it is accumulated up to its excretion carrying out an antioxidant activity [83,95].

Kano et al. (2016) [96] identified the urinary metabolites detected in Wistar rats 4 h after intravenous administration of HTyr and OLE (10 mg/kg and 10 mg/kg, respectively) and oral administration (100 mg/kg and 300 mg/kg, respectively), detecting after oral administration the presence of HTyr, homovanillyl alcohol, homovanillic acid at 1.14, 0.17, 8.03% of the administered dose, respectively, in the urine samples. However, OLE was detected at very low levels (0.07% of the dose) after oral treatment. In contrast, intravenous administration revealed that HTyr and its metabolites were extensively found in the urine while OLE was not detected [96].

Further in vivo studies conducted by Domínguez-Perles et al. (2017) suggest a gender-dependent excretory relationship [81]. The study involved the oral administration of two doses (1 and 5 mg/kg) of HTyr and HTyr acetate on male and female rats. At low doses, the analysis of the metabolites present in the urine revealed that females showed a greater excretion of HTyr derivatives than males, while at high doses of HTyr and HTyr acetate, either females or males showed high levels of HTyr in the urine due to a saturation of Phase I enzymes and intestinal transporters; consequently, the assimilable portion was reduced, and for this reason the presence of compounds in the urine was verified [81].

### 3.4. Toxicity

There is currently great interest in using pure HTyr and OLE as nutraceuticals to prevent cancer, diabetes, chronic and neurodegenerative diseases, and age-related diseases [19]. Several studies have been conducted in cell lines and animal models to identify the possible toxicity of these compounds, but there is little or no literature data on humans. This lack of information currently limits clinical studies.

The acute toxicity, teratogenicity, and mutagenic effects of HTyr have been studied by administering to rats high concentrations of this compound [97,98,99]. A study involving the administration by gastric probe of 2 g/kg of olive pulp containing 70% of HTyr showed no mortality and morbidity effect except for the presence of diarrheal feces. The study allowed to quantify the median lethal dose (LD_50_) at the value of 3.5 g/kg of pure HTyr [98]. However, the toxicological effects were analyzed by oral administration with increasing doses of HTyr (5, 50, 500 mg/kg/day) for 13 weeks, demonstrating that no adverse effects were detected at these doses.

Additionally, no genotoxic and mutagenic effects occur following the administration of HTyr on in vitro models [97]. A particular interest is given by the studies related to the pro-apoptotic and antiproliferative activity on tumor cell lines when treated with pure HTyr or in the form of extract. Several studies have revealed that HTyr is able to inhibit the growth of tumor cell lines, such as colon carcinoma (HT-29, HCT-116, CT-26) [100,101], breast adenocarcinoma (MCF-7), human leukemia (HL-60) [102,103], and pancreatic carcinoma (MIA PaCa-2) [104].

At the same time several studies demonstrated how HTyr exerts selectivity on human non-tumor cell lines [59,105]. Some authors tested HTyr-rich extracts from olive oil mill wastewater, demonstrating that the extracts exhibit antiproliferative and pro-apoptotic effects on the human HL60 cancer cell line and avoids toxicity in the non-tumor cell line [106]. A recent study confirmed the induction of apoptotic and antiproliferative activity on SH-SY5Y neuroblastoma cells, comparing the effects with the non-tumoral cell line MCF10A [107]. The experimental results showed that the highest apoptosis rate was found for HTyr-treated neuroblastoma cells, while the absence of the apoptotic process induction was highlighted for the normal cell line. In vitro studies have shown that OLE has significant activity on several types of cancer cells (colon, liver, prostate, cervix, pancreas, thyroid, lung, leukemia, and neuroblastoma) [108,109]. In addition, some experiments were conducted in animal models to evaluate the antitumor potential in vivo. Encouraging results have been obtained on colon, sarcoma, and skin cancer [108]. Based on the presented investigation and the beneficial effects on health, the European Food Safety Authority (EFSA) has approved since 2011 the health claim on the polyphenols of EVOO, recommending the daily consumption of 5 mg of HTyr and derivatives supplied by 20 g of EVO oil to maintain a balanced diet. This turns out to be the sufficient dose to ensure beneficial health properties by reducing the oxidation of low-density lipoproteins (LDL) and avoiding pro-inflammatory processes.

## 4. HTyr as a Brain Dopamine Metabolite

HTyr can cross the blood–brain barrier, giving rise to a dopamine metabolite and a precursor of oxidized derivatives 3,4-dihydroxyphenylacetaldehyde (DOPAL) and 3,4-dihydroxyphenylacetic acid (DOPAC) [110]. These compounds are also involved in the dopamine biosynthetic pathway and are produced also by HTyr phase I exogen metabolism (Figure 3). They are considered non-specific, being naturally produced through the dopamine and tyrosine endogenous pathways. In fact, the alcohol dehydrogenase (ADH) enzyme catalyzes the oxidation of HTyr into DOPAL while the aldehyde dehydrogenase (ALDH) enzyme converts DOPAL into 3,4-dihydroxyphenylacetic acid (DOPAC), which can produce HTyr by DOPAC reductase [39,83,111].

Other in vivo studies have demonstrated that increased levels of HTyr in urine appeared when red wine is administered, probably due to the interaction between ethanol and dopaminergic pathways. In fact, the authors demonstrated that dopamine metabolism can produce HTyr by reduction of DOPAL catalyzed by aldehyde reductase (ALR); the amount of HTyr in the body can be quantified [112]. The fact that HTyr is also found in brain tissue suggests a role as a dopaminergic neuronal protector [113] being able to increase the antioxidant defenses and carry out its free radical’s scavenger ability [114].

In a double-blind clinical study, the ethanol dose effect on HTyr production in humans was determined by administering ethanol and placebo to twenty-four healthy male volunteers [112]. By testing six different doses of ethanol (6, 12, 18, 24, 30, and 42 g), the urinary excretion of HTyr was evaluated (from 0 to 6 h after administration). The amount of HTyr excreted increased with increasing doses of ethanol. Furthermore, in the same study, a reduction in the DOPAC/HTyr ratio was observed from placebo at the highest dose, consistent with the shift of dopamine metabolism in which HTyr rather than DOPAC was produced [112]. The reductive environment generated during the metabolism of alcohol is responsible for the metabolic change, by promoting the formation of the alcohol derivative (HTyr) instead of the acidic derivative DOPAC [88]. It should be noted that HTyr is the only olive compound that has been found to be a dopaminergic neuronal protector, not only due to its ability to cross the blood–brain barrier, but also due to its potential interactions with dopaminergic pathways [83,115,116].

In summary, differentiating the exogenous and endogenous sources of HTyr in the body becomes a difficult task owing to a wide distribution of HTyr, its short plasma half-life and its possible interaction with different ways such as the dopamine pathway [117].

Since exogenous HTyr is able to cross the brain barrier, as mentioned above, we will summarize its antioxidant and anti-inflammatory effects in nervous tissues and related pathologies associated to the process of aging. In fact, oxidative stress and inflammation in the brain and peripheral nervous system play a pivotal role in the onset and/or progression of neurodegenerative pathologies and neuropathies, as well as in the unbalance of neurogenesis and depression.

## 5. Neuroinflammation in Neurodegenerative Diseases

Neuroinflammation is a hallmark of neurodegenerative diseases such as Alzheimer’s disease (AD) and Parkinson’s disease (PD), where it is prompted by the amyloid-β- and α-synuclein-induced activation of NOD-like receptor family pyrin domain containing 3 (NLRP3) inflammasome, which promotes neurodegeneration [118,119]. The distinctive feature of brain neuroinflammation is microglial activation that occurs following several stimuli, including lipopolysaccharides (LPS), pesticides, beta-amyloid peptide, accumulation of α-synuclein, air pollution, or even neuron damage. Once activated, microglia produce cytotoxic factors, such as superoxide radical (i.e., ROS), nitric oxide (NO), tumor necrosis factor-alpha (TNF-α), and inflammatory prostaglandins [1,120]. In turn, these factors cause cellular modifications that include lipid peroxidation, mitochondrial dysfunction, and altered energy metabolism [121]. ROS and reactive nitrogen species (RNS) are generated mainly in mitochondria as a product of aerobic metabolism. Superoxide radical, hydrogen peroxide (H_2_O_2_), NO, and peroxynitrite are a few of the radical species that are produced as a result of the incomplete reduction of oxygen and nitrogen.

An organism’s lifetime production of ROS and RNS can result in oxidative damage to proteins, membranes, and DNA as well as limit mitochondria’s capacity to create ATP and carry out metabolic processes [1]. The mitophagic efficiency, which is the ability to eliminate less functional mitochondria, decreases with age; in fact, during aging, mitochondria increase in size and decrease in number [122]. A more detailed correlation between neuroinflammation and neurodegenerative diseases, such as PD and AD, is described below in the dedicated sections.

### Effects of HTyr and OLE on Neurodegeneration Connected to Mitophagic Efficiency

Notably, both HTyr and OLE have the ability to increase mitophagy by upregulating mitophagy markers including Beclin, SIRT1, and LC3-II, while downregulating S6 kinase 1 (S6K1) and the AKT/mTOR kinases [123]. Moreover, HTyr and other olive polyphenols are capable of increasing ATP levels in a cellular model of AD (SH-SY5Y-APP695 cells), while the olive-derived secoiridoid ligstroside also improves mitochondrial respiration and biogenesis, extending the lifespan in aged mice and improving the cognitive function [124]. Additionally, treatment with EVOO to 3x-transgenic mice model of AD improved memory performance and produced a significant reduction in insoluble Aβ peptide levels and deposition, which was connected to autophagy activation [125].

Similarly, aged mice (12-month-old), which showed mitochondrial dysfunction evidenced by low brain levels of ATP, NADH-reductase, cytochrome-c-oxidase, and citrate synthase, as well as low expression of SIRT1, cAMP response element-binding protein (CREB), growth-associated protein 43 (Gap43), and glutathione peroxidase 1 (GPX-1), when treated with a mix of purified olive secoiridoids including OLE and HTyr, presented restored brain ATP levels and improved spatial working memory [126]. Thus, mitochondrial function, aging and neurodegeneration are correlated events that can be reverted by HTyr and olive oil polyphenols, also through increase of mitophagy (see Table 1).

## 6. Neuroprotective Function of HTyr and OLE in Neurodegenerative Diseases

### 6.1. Neuroprotective Function of HTyr and OLE in Alzheimer’s Disease

AD is a genetic and sporadic neurodegenerative disease that is a common cause of cognitive impairment acquired in mid- and late-life. AD is characterized by the formation of senile plaques and neurofibrillary tangles (NFT), constituted by extracellular deposit of β-amyloid (Aβ peptide) and by Tau protein accumulation, respectively [163]. In fact, the primary component of NFT are hyperphosphorylated Tau protein isoforms. The physiological function of Tau is to bind to microtubules, thus promoting their formation and stabilization. Microtubules are the main structural component of neurons and are responsible for axonal transport and axon growth. However, hyperphosphorylated Tau is unable to bind microtubules, whose stability is thus reduced, with consequent disruption of cellular traffic and synapse loss of function, cognitive decline, and dementia [163]. Several processes concur to the hyperphosphorylation of Tau, including Aβ peptides, impaired glucose metabolism, and inflammation [164]. Additionally, Aβ peptides are one of the main components of senile plaques and key responsible for AD pathogenesis; Aβ peptides derive from the amyloidogenic metabolism of amyloid precursor protein (APP), which is present in neurons and glia as well as in other tissues. The cleavage of APP by γ-secretase generates different peptides, of which Aβ (1–42) being hydrophobic tends to aggregate in plaques [165].

A remarkable effect of HTyr and OLE (of which HTyr is the main metabolite) is that they inhibit the fibrillization of Tau protein, thus decreasing the intraneuronal and glial lesions [127] (Daccache et al., 2011). Moreover, also β-amyloid aggregation is prevented by HTyr and OLE, since Aβ1–42 oligomer formation was shown to be inhibited in SH-SY5Y neuroblastoma cells [128,166] as well as in a mice model of Aβ deposition (adult TgCRND8 mice), where HTyr restored spatial and associative memory deficit [130]. Equivalent results were obtained after OLE administration in middle-aged TgCRND8 mice [131,132]. There is also evidence that treatment with HTyr reverted the deficit of spatial and working memory induced by intracerebroventricular injection in mice of Aβ1–42 oligomer and also prevented the activation of apoptotic pathways [129]. Reduced β-amyloid aggregation and increased lifespan by OLE was also observed in a transgenic *C. elegans* model expressing Aβ42 [133]. It is worth noting that OLE promotes insulin secretion in pancreatic β-cells and also prevents deposition of amylin in these cells through its HTyr metabolite [134]; similarly, in SH-SY5Y neuroblastoma cells OLE prevents the deposition of α-synuclein, another molecule in the path of β-amyloid aggregation [135]. Moreover, in the neural PC12 cell line, HTyr has been found to prevent the abnormal assembly of α-synuclein and to increase the expression of SIRT-2 deacetylase [136]. However, there is also evidence that the treatment of APP/Ps1 mice (a mouse model of AD) with a low dose of HTyr (5 mg/kg/day) for 6 months does not reduce Aβ deposition, although it is able to reduce mitochondrial protein oxidation and brain inflammation [137]. In comparison, the strong effects on Aβ deposition observed by Nardiello et al. (2018) were obtained with a shorter treatment at much higher dosage of HTyr, which suggests that the dosage is a key variable for effect, more than time [130]. Qin et al. (2021) showed also that HTyr treatment in the APP/PS1 mouse model significantly improved spatial memory in the Morris water maze test, reducing apoptosis in cortex and hippocampus [138]. Interestingly, this neuroprotective effect was dependent on the presence of estrogen receptor β (Erβ), which plays an important role as a neuroprotective agent [138]; in this regard, about two-thirds of AD patients are postmenopausal women.

As mentioned in the Section 5, mitochondrial energetic deficit and altered energy metabolism are features of AD, and it turns out that HTyr is able to acutely induce mitochondria generation and fusion, in a model of Aβ toxicity and mitochondrial dysfunction (cell line 7PA2) [139]. Moreover, it has been shown that OLE and HTyr activate autophagy in SH-SY5Y cells exposed to toxic Aβ1–42 oligomers, thus preventing the accumulation of ROS and the impairment of mitochondrial function [140]. This is relevant considering that autophagy is the main cytoplasmic mechanism of removal and recycling of dysfunctional cellular components, and that the autophagic process decreases during aging, leading to the accumulation of toxic protein aggregates that can trigger neurodegeneration [167].

Another piece of evidence is a report showing that in N2a neuroblastoma cells HTyr or tyrosol counteracted cell death and the activation of NF-kB by nuclear translocation induced by treatment with Aβ25–35 oligomers. However, HTyr did not revert the decrease of glutathione (GSH) triggered by H_2_O_2_ or by Aβ [141]. Furthermore, HTyr neuroprotects astrocytes (C6 cell line) treated with Aβ25–35 oligomers by restoring insulin-signaling; this finding has neuropathological relevance, since astrocytes convert glucose into lactate, which is released into neurons for their functioning, and in fact insulin resistance occurring in diabetes doubles the risk of AD [142]. See Table 1.

### 6.2. Neuroprotective Function of HTyr and OLE in Parkinson’s Disease

PD is a 90% sporadic neurodegenerative pathology, affecting 1–2% of the population over the age of 65, characterized by a specific loss of dopaminergic neurons in the substantia nigra pars compacta [168]. PD is characterized by neuropsychiatric symptoms such as depression and anxiety that precede the onset of motor symptoms, consisting of resting tremors, muscular rigidity, and postural instability. Neuropathological hallmark is the presence of an intracytoplasmic inclusion body, known as a Lewy body, generated chiefly by abnormal aggregation of α-synuclein in substantia nigra, locus ceruleus, nucleus basalis, and hypothalamus [169]. α-synuclein, which is also a component of AD amyloid plaques (see above), is a presynaptic nerve terminal protein whose physiological function is neuroprotective and pro-neurogenic [170]. There is evidence that PD pathogenesis is associated to oxidative stress caused by ROS generated by a mitochondrial dysfunction in the respiratory chain [171]. It appears that in PD, etiology plays a role in the oxidation of dopamine by monoamine oxidase (MAO) in substantia nigra neurons, where unstable oxidized dopamine molecules (quinones) cause mitochondria dysfunction, formation of neurotoxic a-synuclein fibrils, dysfunction of the lysosomal system, and oxidative stress. ROS, in turn, activate microglial cells that promote neuroinflammation, causing death of dopaminergic neurons [172,173]. There are therefore several inter-related factors causing PD, while no specific therapy halting the neurodegenerative progression has been identified.

In this context, HTyr butyrate has been shown to inhibit the apoptosis induced in SH-SY5Y cells by the PD-related neurotoxin 6-hydroxydopamine (6-OHDA) through activation of the Nrf2/heme oxygenase-1 axis (Nrf2/HO-1) [143]. Nrf2 is a transcription factor that plays a key role in the protection against oxidative stress, as it regulates a large number of antioxidant and anti-inflammatory genes and also modulates species longevity [174]. HTyr butyrate acts by inhibiting the ubiquitin E3 ligase that targets Nrf2 for proteasomal degradation, after reacting with the cysteine thiols of Kelch-like ECH-associated protein 1 (Keap1) that is part of the ubiquitin complex [143,175]. Consequently, Nrf2 is stabilized and accumulates in the nucleus where it induces the antioxidant enzyme HO-1 that, in turn, prevents cell damage by various types of ROS-dependent oxidative stresses [174]. Notably, Peng et al. (2015) [144] demonstrated that in PC12 cells the HTyr treatment causes a rapid increase of nuclear Nrf2 protein levels, which protects from death caused by H_2_O_2_ or 6-OHDA toxicity (see also below). It is worth noting that also in non-neural systems, HTyr has been found to activate HO-1 by stabilizing Nrf2, for example as part of a wound healing process (tested in vascular endothelial cell culture) or of a protective action against LPS-induced inflammation [176,177]. Moreover, in the Nrf2 and HO-1 activation by HTyr, the phosphatidylinositol 3-Kinase (PI3K)/Akt and extracellular signal-regulated kinase (ERK1/2) signaling pathways are also implicated, as assessed using specific inhibitors [176]; while the Janus kinase/Signal transducer and activator of transcription (Jak/Stat) pathway is inhibited by paracetylated HTyr [177]. As further example of Nrf2 induction by HTyr outside the nervous system, HTyr increased mRNA expression of Nrf2 and its downstream genes in the small intestine of diquat-challenged mice [178].

Additionally, OLE is able to neuroprotect from death PC12 cells exposed to 6-OHDA by reducing mitochondrial production of ROS and favoring autophagy [145]. In another work, Yu et al. (2016a) showed that treatment with HTyr protected from death SH-SY5Y cells exposed to dopamine or to 6-OHDA [146]. The authors proposed that this effect depends not only on HTyr antioxidant activity but also on its ability to induce phase II enzymes, including glutathione S-transferase (GST), HO-1, and NADPH quinone oxidoreductase 1 (NQO1), which lessens quinones’ toxicity by reducing them. Another interesting work shows that in PC12 cells, HTyr is able to inhibit the production of the toxic dopamine metabolite 3,4-dihydroxyphenylacetaldehyde (DOPAL), generated in great amount from endogenous dopamine after inhibition of monoamine oxidase (MAO) through specific MAO inhibitors [147]. Therefore, HTyr was able to inhibit the oxidation of dopamine, either enzymatic or spontaneous, and this is relevant if we consider that MAO inhibitors are used for PD therapy. Another report shows that HTyr is able to acutely exert an antioxidant effect by inhibiting MAO enzymes after injection in mice striatum of 1-methyl-4-phenylpyridinium (MPP+), a potent oxidant [148]. As mentioned above, a theory for the etiology of PD is that the disease is caused by the increase of nigrostriatal dopamine catabolism by MAO enzymes, thus inducing a rise in ROS levels and death of dopaminergic neurons. A further report by the same authors [149] shows that pretreatment of rats with HTyr or with HTyr acetate or with nitro-HTyr 5 min before intrastriatal infusion of MPP+, protected from dopamine neuron degeneration, as ipsilateral turns were decreased and the GSH/GSSG ratio (glutathione/glutathione disulfide) was increased. Since GSSG is produced by antioxidant enzymes from GSH during the reduction of ROS, this indicates that the oxidative stress in the striatum was reduced.

Additionally, OLE has been shown to be neuroprotective in a mouse model of PD induced by Rotenone, which inhibits the mitochondrial complex I causing overproduction of ROS and aggregation of α-synuclein. OLE (16 to 32 mg/kg) regulates the brain-derived neurotrophic factor (BDNF)/CREB/Akt signaling pathway, restoring BDNF levels and Akt phosphorylation and reducing aggregation of α-synuclein [150].

Remarkably, a biochemical study by Palazzi et al. (2020), conducted in the SH-SY5Y cell line, shows that HTyr did not change the naturally unfolded structure of α-synuclein but stabilizes certain areas of the molecule, thus preventing protein fibrillation [151]. Similarly, OLE was found to reduce the toxicity of α-synuclein aggregates favoring monomerization [152]. A further conformational study showed that OLE interacts with the *N*-terminal domain of α-synuclein, making it unable to react with cellular lipid membranes and thus preventing the generation of toxic aggregates [153]. Other studies in *C. elegans* showed how HTyr and OLE are able to increase lifespan in normal conditions or after heat stress and prevent α-synuclein aggregation, while in a PD model, HTyr increases locomotion and prevents α-synuclein accumulation in dopaminergic cells and in muscle, thus reducing neurodegeneration [154,155]. See Table 1.

### 6.3. Effects of HTyr and OLE in Diabetes-Induced Neurodegeneration

Type 1 and type 2 diabetes arise from a failure in insulin production or from resistance to insulin action, respectively. Insulin metabolism plays a major role in the onset of neurodegenerative diseases, in particular AD. Type 2 diabetes-induced hyperglycemia causes endothelial cells, pericytes, and astrocytes to enhance their mitochondrial respiration, which boosts ROS generation and oxidative stress. In fact, insulin receptors (Irs) are present in the brain, maximally in the cerebral cortex and hippocampus, and their ligands are insulin, IGF-1, and IGF-2 [179]. Through insulin and IGFs, Irs regulate the metabolism of glucose in the brain and exert inhibitory effects on apoptosis in brain neuronal cells [180]. The majority of cellular and molecular processes, including protein synthesis, sorting, transport, and degradation, as well as the maintenance of synaptic transmission, all require ATP, which can only be produced from glucose. Noradrenaline and/or cortisol, whose levels both increase with advancing age, can impede receptor activity at several locations, leading to desensitization of the neuronal insulin receptor in late-onset sporadic AD disease. Crucially, unregulated glucose metabolism causes AD symptoms [181]. In fact, insulin and amyloid-β compete for binding to IR. Insulin resistance is the outcome of this competition, which lowers the affinity of insulin to IR; thus, insulin resistance together with neuroinflammation and oxidative stress are causes of amyloid-β toxicity and accumulation of neurofibrillary tangles [182].

HTyr treatment of *db*/*db* mice, a model of type 2 diabetes, induces the Nrf2/HO-1 pathway in the brain and activates the PGC-1α transcriptional coactivator, AMPK kinase, and the deacetylase SIRT1 [156], which are part of an energy sensing network [183]. In fact, PGC1-α is, as mentioned in the Introduction, a master regulator of mitochondrial biogenesis and is directly regulated by the two metabolic sensors AMPK and SIRT1, through phosphorylation and deacetylation, respectively [183]. Thus, HTyr improves mitochondrial function and prevents oxidative stress in the *db*/*db* mice brain by activating the SIRT1/AMPK/PGC1α axis. Moreover, the amyloid protein (islet APP) generated in the diabetes mellitus pathology in β-cells of pancreas share a structure similarity with neuron APP and, consistently, islet APP fibrillation is prevented by HTyr [184].

It is worth noting that the activation of SIRT1 deacetylase by HTyr is functionally significant because SIRT1 exerts multiple actions that extend lifespan and have anti-aging effects [14]. These actions are indicated in the Introduction and include: (1) the inactivation of the pro-aging transcription factor NF-kB by deacetylating residue 310 of the p65/RelA subunit [185]; (2) blockage of the mTOR pathway, which results in rescue of the autophagy defect induced by oxidative stress [14]; notably, suppression of mTOR activity has been shown to increase lifespan in mice and other organisms [186]; (3) activation of PGC1α resulting in downregulation of ROS production mediated by NADPH-oxidase [187]; furthermore, PGC1α exerts an antisenescence action by promoting homeostasis of mitochondria, which, as mentioned above, is crucial to prevent oxidative stress, a condition that increases with aging.

Another interesting study was conducted with OLE utilizing a streptozotocin (STZ)-based diabetic rat model. Authors observed that the outcome of diabetes consisted of reduced performance in spatial memory tests, neuroinflammation symptoms of oxidative stress such as decrease of superoxide dismutase (SOD), and of the pro-inflammatory cytokines IL-1β and TNF-α, together with a decrease of the phosphorylated forms of PI3K, AKT, and mTOR kinases in the hippocampus; all these changes were reverted by treatment with OLE, indicating that neuroinflammation and cognitive dysfunction can be attenuated by OLE [157]. More generally, there are several reports, reviewed by Zheng et al. (2021) [158], indicating that OLE displays multiple actions against diabetes, as it regulates insulin secretion, restores islet morphology, activates AMPK signaling, and ameliorates glucose tolerance and insulin resistance; consequently, OLE relieves diabetes-associated diseases including diabetic nephropathy, cardiovascular complications, and diabetic retinopathy [188]. See Table 1.

### 6.4. Effects of HTyr in Multiple Sclerosis

Multiple sclerosis (MS) is a chronic disease of the central nervous system characterized by loss of sensory and motor function, associated to immune-derived inflammation, demyelination, and consequent axonal damage. The clinical features indicate that MS patients frequently present recurrent episodes (relapses) of neurological disease, while for the majority (70%) the disease becomes chronic and progressive with age. Therefore, although MS is the most common cause of neurological disease in young adults (more than two million worldwide), the pathology aggravates during aging [189]. It is characterized by perivenular inflammatory lesions, caused by infiltrates of T-lymphocytes, leading to demyelinating plaques, which are the hallmark of MS. Inflammation results in oligodendrocyte damage and demyelination [190]. While axons remain almost intact during the early stages of the disease, the progressive evolution is an irreversible axonal damage [191]. The most common syndromes involve optic neuritis, brainstem, and spinal cord, and less frequently, parietal lobe syndromes. The inflammatory process in MS appears to be triggered by an autoimmune action implicating T-cells targeting myelin self-antigens, and to involve cross-reaction to antigens expressed by myelin elements or by viruses [192].

HTyr was shown to be able, in a rat experimental model of MS (experimental autoimmune encephalomyelitis, EAE), to reduce lipid and protein oxidation, and to increase GPX, thus reducing the oxidative stress produced by EAE [159]. Similarly, in primary rat astrocytes activated by LPS, HTyr, in a mix with tyrosol, inhibited matrix metalloproteinase-2 (MMP-2; also known as gelatinase A) and MMP-9 (gelatinase B); these are two proteolytic enzymes involved in inflammatory processes and in MS, since, for instance, MMP-9 increases blood–brain barrier permeabilization to leukocytes infiltration into the CNS [160]. This protective effect was not observed in treatment with the flavonoids quercetin and catechins. Additionally, OLE, administered by olive leaf extract (45 mg/kg) in an EAE rat model, up-regulated GPX1, SOD1 and SOD2 activity, SIRT1, and anti-inflammatory M2 microglia, whereas it downregulated proinflammatory M1, favoring myelin integrity. This indicates that OLE can be used to treat MS [161]. See Table 1.

### 6.5. Effects of HTyr in Huntington’s Disease 

Huntington’s disease (HD) is a mostly hereditary neurodegenerative illness (autosomal dominant) caused by a mutation in the huntingtin protein, which results in an expanded number of the CAG trinucleotide repeat (i.e., over 28 repeats) within a polyglutamine tract in the molecule. Due to the multiple activities of huntingtin and to the cytotoxicity of large amounts of glutamine, this causes damage in basal ganglia (striatum) followed by degeneration in the cortex and hippocampus, accompanied by aberrant motor symptoms and depression [193]. HD onset depends on the extension of the mutation, but symptoms frequently appear during late age and worsen progressively [193].

Tasset et al. (2011) showed that in a rat model of HD, induced by intraperitoneal administration of 3-nitropropionic acid (3NP, a mitochondrial toxin that causes degeneration preferentially in the striatum), the administration of HTyr and EVOO for two weeks exerted antioxidant activity [162]. In fact, the authors observed that HTyr and EVOO reduced lipid peroxidation and reverted the decrease of GSH content in the striatum and the rest of the brain. Moreover, EVOO reverted the blockage effected by 3NP of succinate dehydrogenase, an enzyme abundant in mitochondria that connects the TCA cycle to oxidative phosphorylation of the respiratory chain. Thus, EVOO and Htyr, behaving as powerful antioxidants, represent an encouraging possibility to treat HD, which today has no effective therapy. See Table 1.

## 7. Effects of HTyr in Adult Neurogenesis and in Stroke

Adult neurogenesis is the process by which new neurons are generated throughout life in the brain from neural stem cells, in two neurogenic niches, the subgranular zone of the dentate gyrus of the hippocampus and the subventricular zone [194,195,196]. In the hippocampus, the new neurons are necessary for learning and memory, as they greatly enhance the built-in ability of the dentate gyrus to distinguish between similar memory patterns (pattern separation) [197,198]. The new neurons are continuously generated from stem cells in the subgranular zone of the dentate gyrus, which mature into proliferating progenitor cells and neuroblasts, before becoming post-mitotic neurons. There is debate about whether adult neurogenesis also occurs in humans, as there is evidence both in favor, even in old age [199], and against [200]. Moreover, the stem cell pool undergoes a slow process of self-renewal. Two models of self-renewal have been suggested, one proposing a recurrent self-renewal, where the same stem cell pool produces neurons through repeated rounds of division, remaining available for further activation [201,202,203,204], and another in which stem cells are rapidly depleted after a number of divisions [205]. In the first model, the pool is preserved also in old age. Notably, however, during aging there is a decrease in the generation of new neurons [206], which results in a reduced ability to perform hippocampus-dependent memory tasks [207].

Some neurogenic stimuli, such as physical exercise, can partially revert the decrease of neurogenesis occurring during aging [208,209,210], while other neurogenic stimuli, though effective in the adult, are unable to elicit neurogenesis in aged models. Between them are treatment with antidepressant molecules regulating 5-hydroxytryptamine (5-HT; selective 5-HT reuptake inhibitors, i.e., SSRI) or norepinephrine pathways [211,212,213], and learning [214,215]. Concerning nutrients, we have recently shown that HTyr activates hippocampal neurogenesis, including stem cells, in aged mice [47]. The majority of these neurogenic stimuli activates only progenitor cells and not stem cells, for example running [209] or the antidepressant fluoxetine [216,217,218]. However, some nutrients are also able to activate stem cells, though not in aged mice, but in models that are defective for neurogenesis, as for instance the natural flavonoid luteolin in a mouse model of Down syndrome [219,220]. Additionally, HTyr increases the survival of active (c-fos+) new neurons (see Section 10.3) and decreases the expression of the markers of aging and neuroinflammation lipofuscin and ionized calcium-binding adapter molecule 1 (Iba1) [47]. Overall, the finding that HTyr reactivates aging stem cells supports the idea of using dietary supplements to counteract cognitive deterioration during aging. Another report demonstrated that HTyr treatment of prenatally stressed rat mothers (by restraint stress on days 14–20 of pregnancy) restores in the offspring of treated mothers the mRNA levels of BDNF, GAP43, synaptophysin, and *N*-methyl-D-aspartate (NMDA) receptor subunits NR1, NR2A, and NR2B, all neural markers involved in synaptic plasticity and decreased by stress in the entire hippocampus [221]. However, no information was given regarding the generation of new neurons in the neurogenic niches [221]. More recently, a report about treatment with HTyr in vivo after stroke, obtained with transient occlusion of the middle cerebral artery, showed that HTyr-fed mice presented improved short-term recognition memory and an increase of BDNF and of cerebral blood flow in the hippocampus. Moreover, the study suggested a trend of increase of dentate gyrus total progenitor cells (DCX+ cells), without, however, showing a significant difference between treated and untreated conditions after stroke [222]. This may also depend on the dosage of HTyr treatment, lower than that used in the study by D’Andrea et al. (2020) [47]. This report, together with that by [221], brings to evidence the synaptogenic potential of HTyr, an important feature in the process of neural regeneration occurring after stroke. Additionally, OLE treatment, in a rat model of stroke by occlusion of the middle cerebral artery, induces a decrease of cerebral edema and plasma fibrinogen, inhibition of angiotensin converting enzyme (ACE) activity, and an increase of the antioxidant enzymes SOD, GPX, and catalase in brain tissue [223]. Another previous report in a mouse model of stroke by occlusion of the middle cerebral artery followed by reperfusion indicates that OLE (100 mg/kg i.p.) reduces the volume of cerebral infarction and apoptosis through reduction of Bax and increase of Bcl2 expression [224]. Likewise, OLE treatment after brain stroke caused in rats by intracerebral hemorrage following a collagenase injection in the brain improves brain edema and protects the integrity of the blood–brain barrier [225]. See Table 2.

## 8. Effects of HTyr and OLE in Stress, Anxiety and Depression-like Behavior

Depression is the most common psychiatric illness, affecting more than 350 million people worldwide. Depression is favored by stress, and its etiopathology has been related to defects in hippocampal neurogenesis, as the effect of antidepressants is conditional to their ability to induce neurogenesis in rodents as well as in non-human primates [211,217,218]. Stress and depression are associated with conditions that reduce the ability to generate new hippocampal neurons, such as high-plasma glucocorticoids, which inhibit neurogenesis; in fact, glucocorticoids and corticosterone are produced under stress in the adrenal gland, as part of the hypothalamic-pituitary-adrenal (HPA) axis, and glucocorticoid receptors are abundant in the hippocampus (see for review [235]. Conversely, inducers of hippocampal neurogenesis, such as antidepressants and voluntary exercise, reduce anxiety and depression-like behaviors induced by stress in experimental models, such as restraint, cold, or forced swimming. Indeed, in correlation to its pro-neurogenesis effects, running improves performance in the major tests assessing depression-like behavior (learned helplessness, forced-swim, and tail suspension) and anxiety (elevated plus-maze and open field) [236]. Remarkably, it has been shown that the optogenetic activation of the ventral dentate gyrus in the hippocampus, as detected by c-fos immunoreactivity, leads to a decrease of anxiety in mice, as measured by the plus maze test [237]. However, antidepressants and exercise exert antidepressant activity also when neurogenesis is prevented by chemical or genetic means; furthermore, the depletion of neurogenesis in itself is not sufficient to directly cause anxiety/depression [238], indicating that depression depends also on non-neurogenic factors, such as neural plasticity and blood vessel density. There is also a correlation between depression and aging-related processes, such as neuroinflammation, amyloid accumulation, changes in neuroplasticity, and synaptogenesis, that increase the risk of late-life-depression [239]. It is noteworthy that during aging, the production of new neurons decreases, and the antidepressant fluoxetine is unable to elicit neurogenesis in aged mice, but still has antidepressant activity through its modulation of neuronal plasticity, thus evidencing the importance of this component in depression [240].

In this context, it should be considered a potential antidepressant effect of HTyr, given its ability to stimulate hippocampal neurogenesis and neuron survival in young and aged mice [47].

In fact, a report showed that mice with depression-like behavior induced by chronic unpredictable mild stress (CUMS), after 7 weeks of treatment with HTyr, performed significantly better in forced swimming and tail suspension tests, in association with reduced oxidative stress—through enhanced SOD activity—and with increased number of glial fibrillary acidic protein (GFAP)-immunoreactive astrocytes, as well as increased activity of the BDNF/TrkB/CREB signaling pathway [226]. Considering that BDNF is important for the maturation of new neurons and is increased by neurogenic stimuli such as running and antidepressants [218,241], this report suggests that HTyr acts as antidepressant by stimulating neurogenesis. Another report using the CUMS model, explored the effect of HTyr on the hypothalamic–pituitary–adrenal (HPA) axis [227]. In fact, there is evidence that HPA axis is hyperactive in conditions of stress and in major depressive disorders [242] and contributes to the etiopathology. Fan et al. (2021) found that HTyr has a significant antidepressant effect associated to improvement of the HPA axis, as suggested by the decrease effected by HTyr after CUMS of serum corticosterone, adrenocorticotropic hormone (ACTH), and also TNF-α, IL1β, and IFN-γ [227]. However, they also observed in the hippocampus a strong HTyr-induced rescue of the decreased levels of 5-HT, which is a key player in the onset of depression. In fact, serotonin through hippocampal 5-HT1,3,4,6,7 receptors modulates proliferation [243] and the deletion of 5-HT1a receptor impairs the neurogenic effect of fluoxetine [211]. On the other hand, Fan et al. (2021) showed that HTyr was able to revert only partially the alteration in gut microbiota phyla [227]. Thus, HTyr displays a mixed antidepressant-like effect on the HPA axis and on the hippocampus.

Similarly to HTyr, OLE treatment reduced in rats the anxiety elicited by a single prolonged stress (SPS), such as that induced by immobilization and forced swim tests, as assessed by the plus maze and open field tests that are able to detect the level of anxiety. The induction of a strong and prolonged stress, such as SPS, elicited symptoms similar to post-traumatic stress disorder (PTSD), a stress-related mental disorder caused by a traumatic experience [228]. A possible anti-PTSD mechanism of OLE is the ability to restore in the hippocampus the levels of Neuropeptide Y, which modulates the serotonergic pathways, as well as the levels of BDNF [228]. This is relevant considering the existence of stress-inhibitory pathways in the dentate gyrus activated by neurogenesis [237]. OLE (8 to 32 mg/kg i.p.) was also shown able to counteract depression-like symptoms elicited in mice by daily administration of corticosterone (40 mg/kg, i.p.) for 21 days. In fact, OLE treatment significantly improved the performance in the tail suspension test and in the forced swimming test, and restored brain serotonin and dopamine levels [229] (see Table 2).

## 9. Effects of HTyr and OLE in the Peripheral Nervous System: Nerve Damage, Neuropathies, and Regeneration

Older age increases the likelihood of hyperalgesia, as chronic pain incidence increases with age; in particular, peripheral neuropathic pain occurs in 35% of patients (typically due to diabetes or postherpetic neuralgia), and recovery from peripheral nerve damage takes longer [244].

In a report using a rat model of chronic compression of the dorsal root ganglion (CCD) to reproduce the neuropathic pain caused by intervertebral disc degeneration (IVDD), HTyr (acutely administered intrathecally in the spinal cord) reduced the levels of some inflammatory molecules activated by the NF-kB pathway, i.e., cyclooxygenase-2 (COX-2), NLRP3, nitric oxide synthase (iNOS), and metalloproteinase with thrombospondin motifs-4 (ADAMTS-4) [230].

Interestingly, another report demonstrated that HTyr (20 ng/mL) stimulates the proliferation of primary human Schwann cells as well as the protein expression of GFAP and p75 nerve growth factor receptor (p75 NGFR) [231]. Given that Schwann cells are implicated in peripheral nerve biology, including nerve formation and regeneration, transmission of nervous impulses along axons, and trophic support for neurons, this implies that the HTyr neuroprotective effects may facilitate regeneration and nerve trophism.

Consistently, HTyr orally administered for 6 weeks (100 mg/Kg) was able to reduce peripheral neuropathy in STZ-treated diabetic rats [232]. In fact, HTyr counteracted the decrease of tail nerve conduction velocity and the increase of thermal nociceptive threshold occurring after STZ treatment. HTyr also abolished the decrease of Na+K+ ATPase activity in the sciatic nerve [232] indicating, as a whole, that HTyr may be a suitable therapy for the early-stage diabetic neuropathy.

Additionally, OLE displays beneficial effects on inflammation ensuing spinal cord injury (SCI), since it was shown to reduce neutrophil infiltration, which plays a critical role in post-traumatic inflammation, as these cells cause large secondary tissue damage. Myeloperoxidase activity, a unique biomarker of the amount of neutrophil infiltration, was measured in order to determine if post-traumatic neutrophil infiltration had decreased [233].

HTyr treatment in vivo of rats (10 mg/Kg) also mitigated spinal cord injury after laminectomy, since the neural function was rescued and lipid peroxidation and myeloperoxidase activities were reduced, together with decrease of proinflammatory cytokines and apoptotic markers [234] (see Table 2).

## 10. Neuroprotective Effects Exerted by HTyr In Vitro, In Vivo, and in Retinopathies

The neuroprotective effect of HTyr is one of the most marked properties of this molecule and is also relevant when viewed as a safeguard against age-dependent neurodegeneration. We have already discussed about neuroprotection in neurodegenerative diseases (see above). There are however several examples of neuroprotection by HTyr in other systems, which we report in this section.

### 10.1. Neuroprotective and Anti-Inflammatory Effect in Cells Lines and Microglia

In brain-dissociated cells exposed to NO cytotoxicity, an HTyr-rich olive extract prevented ATP loss, mitochondrial membrane depolarization and lipid peroxidation [245]. Brain cells were obtained from mice pre-treated with 100 mg/kg HTyr for 12 days. Moreover, following LPS-induced inflammation in the BV2 microglia cell line and in primary microglia cells, HTyr reduced the expression of the M1 macrophage pro-inflammatory mediator CD86 while increasing the M2 macrophage anti-inflammatory marker CD206; it also reduced the phospho-NF-kB p65 protein and repressed the toll-like receptor 4 (TLR4) [246]. All this may contribute to reduce the microglia inflammation, a key player in the inflammatory progression of neurodegeneration (see Section 5). Another report points to the anti-inflammatory role of HTyr in microglia, using the BV2 cell line activated with LPS or α-synuclein. The authors showed that HTyr is able to reduce the translocation—and the consequent activation—of NF-kB to the nucleus elicited by LPS, but not the translocation elicited by α-synuclein. Conversely, they observed that NADP oxidase, which is the main source of ROS in microglia, is reduced by HTyr after α-synuclein-mediated activation of microglia, but not significantly after activation with LPS [247]; similarly, the authors showed that LPS but not α-synuclein induced the expression of the inflammasome NLRP3 and that HTyr was able to reduce NLRP3 levels [247]. This, indicates that HTyr has different anti-inflammatory mechanisms versus microglia.

In PC12 cells treated with H_2_O_2_ or with 6-OHDA, HTyr completely protected from death via activation of the Keap1-Nrf2 pathway, upregulating nuclear Nrf2 and its targets, i.e., cytoprotective enzymes such as glutamate–cysteine ligase (GCLC, part of the GSH production system), HO-1 (part of the iron metabolism), NQO1 (part of the ROS detoxification system), and thioredoxin reductase (TXNRD1, key detoxificant molecule) [144]. This direct effect of HTyr on Nrf2 transcription factor agrees with the findings by Funakohi-Tago et al. (2018) [173], and is remarkable since, as mentioned in the Section 6.2, Nrf2 is a key cytoprotective molecule that regulates GSH production, anti-ROS molecules, the thioredoxin (TXN) antioxidant system, NADPH regeneration, and heme metabolism (see for review [248]. Similarly, a two-fold increase of cell viability occurred in PC12 cells overexpressing synapsin1 and with downregulation of septin-5 (synaptic vesicle proteins), treated with 100 μM HTyr, under oxidative stress triggered by salsolinol, a dopaminergic neurotoxin. This effect was accompanied by an increase of catalase (CAT), glutathione reductase (GR) and peroxidase (GPX), and SOD [249]. All these enzymes are antioxidant, since the GPX function is to reduce lipid hydroperoxides to alcohols and H_2_O_2_ to water, CAT as well reduces H_2_O_2_ to H_2_O, while GR catalyzes the formation of antioxidant GSH from GSSG, and SOD degrades superoxide molecules (O−2). Thus, the HTyr-mediated activation of Nrf2, the master regulator of anti-oxidant defense, may be at the basis of the majority of the cytoprotective actions exerted by HTyr (see Figure 4 and Table 3).

The figure depicts the activation by HTyr of Nrf2, which regulates several anti-oxidizing systems, and can thus be representative of the main antioxidant activity of HTyr. In red are the molecules demonstrated to be up- or down-regulated (NOX2) by HTyr [143,144,248,249,263,264].

The HTyr/Nrf2-activated ROS detoxification system includes glutathione peroxidase 2 (GPX2), which uses GSH to transform hydrogen peroxide (H_2_O_2_) in H2O; NADPH quinone dehydrogenase 1 (NQO1) reduces quinone to hydroquinone using nicotinamide adenine dinucleotide phosphate (NADPH); catalase (CAT) converts H_2_O_2_ into water; superoxide dismutase 2 (SOD2) transforms hydroperoxyl (HO_2_, i.e., the anion superoxide O2− in aqueous solution) into oxygen and hydrogen peroxide, which is less toxic than superoxide [156]. Finally, NADPH oxidase (NOX2), which produces pro-inflammatory superoxide from oxygen and NADPH, is negatively regulated by Nrf2 [265] and by HTyr [263].

Part of the GSH production system is the heterodimer glutamate–cysteine ligase catalytic subunit (GCLC)/glutamate–cysteine ligase modifier (GCLM), responsible for the first step of GSH production from L-glutamate and L-cysteine) and glutathione synthetase (GS), responsible for the second step leading to the production of GSH. Moreover, Solute Carrier Family 7 Member 11 (Slc7A11 or xCT; cystine transporter) and glutathione reductase (GR) (see also panel on the left) contribute to GSH production.

The thioredoxin (TXN) system is responsible for the reduction by TXN1 of oxidized thiol groups in proteins and lipids, and TXN1 oxidized in this process is reconstituted to reduced TXN1 by thioredoxin reductase (TXNRD1) through the concomitant oxidation of NADPH.

NADPH is generated by several enzymes all induced by Nrf2, i.e., glucose-6-phosphate dehydrogenase (G6PD), 6-phosphogluconate dehydrogenase (6PGD), isocitrate dehydrogenase 1 (Idh1), and malic enzyme 1 (Me1). G6PD reduces NADP^+^ to NADPH while oxidizing glucose-6-phosphate (G6P) to 6-phosphogluconolactone (6PG).

HMOX (heme oxygenase, also known as HO-1) is part of the iron metabolism together with the ferritin complex. HMOX1 breaks heme into biliverdin and Fe^2+^, which catalyzes the generation of the damaging OH free radical from H_2_O_2_ through the Fenton reaction; however, the ferritin complex, acting in parallel with HMOX1, inactivates Fe^2+^ by incorporating it and modifying into Fe^3+^ (see also [176] for induction of HO-1 by HTyr; moreover, HTyr reduces the increase of ferritin observed after H_2_O_2_ treatment [266].

### 10.2. Ex-Vivo Experiments of Hypoxia-Reoxygenation

HTyr treatment has been widely tested in hypoxia-reoxygenation experiments conducted in brain slices. This is an in vitro model of ischemia used to study the recovery process after a hypoxic period, which is accompanied by ROS increase. This model can be relevant in predicting possible effects of HTyr and other antioxidant molecules on aging, since intermittent hypoxia can be paradoxically beneficial in aging and in neurodegenerative diseases and could potentially modulate brain aging [267]. In fact, the antioxidant enzyme basal activity (i.e., SOD, CAT, or GPX activity) is not affected by aging or is even increased, but in the hypoxia-reoxygenation model the post-hypoxic response to the increased ROS is reduced [268]; thus, the antioxidant effects of HTyr observed in this model can be translated in beneficial effects during aging.

In diabetic rat brain slices undergoing hypoxia-reoxygenation, HTyr reduces lipid peroxidation as well as NO and peroxynitrite concentration [250]. In another example of hypoxia-reoxygenation of brain slices from normal rats, HTyr reduced cell death, as measured by lactate dehydrogenase (LDH) efflux to the incubation medium [251]. Similarly, the increase of the inflammatory prostaglandin E2 and IL-1β induced by reoxygenation in brain slices was inhibited by HTyr treatment directly on slices or after oral administration (10 mg/Kg/day) [252]. Another study aimed to detect possible synergies between polyphenols in rat brain slices tested the effect of HTyr alone or together with tyrosol, dihydroxyphenylglycol, and oleocanthal on cell survival (LDH efflux), or against oxidative stress (lipid peroxidation) and nitrosative stress (peroxynitrite production); it turned out that HTyr had cytoprotective and antioxidant effects that were potentiated by dihydroxyphenylglycol and oleocanthal [253]. Furthermore, a comparative analysis measuring the protective effect of the catechol group against lipid peroxidation caused by ferrous salts, depletion of GSH induced by diethylmaleate or by hypoxia-reoxygenation, or cell death induced by hypoxia-reoxygenation, evidenced how HTyr ethyl ether (with two OH groups) was more effective than tyrosol ethyl ether (one OH) or 3,4-di-*ortho*-methylidene-hydroxytyrosol ethyl ether (without free OH); [254]. See Table 3.

### 10.3. In Vivo Neuroprotection by HTyr

In vivo treatment of aged and young mice with HTyr administered in drinking water (100 mg/kg) for one month was shown to result in neuroprotection, as indicated by increased survival of hippocampal neurons, and decreased apoptosis. Furthermore, the new neurons were functionally integrated into neuronal circuits, as indicated by increase of *c-fos* expression; an increase of dentate gyrus stem cells (Sox2-positive) was also observed, as well as a decrease of the aging markers lipofuscin and Iba-1, which indicates a rejuvenating action of HTyr [47]. Another report, already mentioned in Section 8***,*** shows how HTyr reduced the depression-like behaviors induced in mice by exposure to CUMS. In fact, HTyr increased SOD activity, reduced ROS production, and inhibited the activation of microglia (by reducing TNF-α and IL1-β), thus effectively reducing oxidative stress. Additionally, HTyr increased the number of hippocampal astrocytes positive for GFAP, an indication of enhanced neurogenesis [226], in agreement with the findings by D’Andrea et al. (2020) [47].

Another report showed that in a rat model of brain trauma by subarachnoid hemorrhage (SAH), induced through perforation of the carotid artery, HTyr treatment (100 mg/Kg per day, for 6 weeks) caused a reduction of NF-kB levels and had antioxidant effects by restoring the levels of the antioxidant enzymes SOD, GPX, and CAT. Furthermore, HTyr did not show effects on mortality rate, but caspase-3 activity was reduced in the cortex [255]. See Table 3.

### 10.4. Neuroprotection from Retinopathy by HTyr and OLE

In diabetes, hyperglycemia is associated with the alteration of small blood vessels and intracellular endothelial oxidative stress, together with neurotoxicity and ischemia in neural tissues, including brain, peripheral nerves, and retina [269].

In rats with diabetes induced by STZ, the treatment with HTyr started 7 days before inducing diabetes and continuing for 2 months after (5 mg/Kg/day by gavage) blunted the decrease of retinal ganglion cells, thus exerting a neuroprotective effect [256]. Similarly, HTyr was neuroprotective in a cellular model of smoking- and age-related macular degeneration (AMD), i.e., the human retinal pigment epithelial cell line, ARPE-19, exposed to acrolein, a component of cigarette smoke that reduces cell viability and causes oxidative damage [257]. In fact, pretreatment with HTyr protected ARPE-19 cells from oxidative damage and mitochondrial dysfunction induced by acrolein [257]. Interestingly, the effect of HTyr (in a formulation together with resveratrol, and the carotenoids zeaxanthin and lutein) in humans aged 50 years or older diagnosed with unilateral exudative AMD, was analyzed through a multicenter trial. AMD is associated with progressive damage to macula, i.e., the central part of retina, and is a leading cause of loss of vision, and no effective therapy is available today. It turned out that after 12 months of treatment with HTyr, *n*-6 polyunsaturated fatty acids (*n*-6 PUFA) increased, while inflammatory cytokines IL-8, IL-1β, and TNF-α were significantly reduced, indicating a decrease of the inflammatory condition; however, no difference in visual acuity was noted [258]. OLE can also reduce diabetes complications including diabetic retinopathy. In fact, Benlarbi et al. (2020) observed that in type 2 diabetes model *Meriones shawi*, treatment with oleuropein protected retinal cells in culture against the toxic effect of glucose by improving the viability of photoreceptors [188]. See Table 3.

### 10.5. Neuroprotective Effect of HTyr Derivatives

Several HTyr derivatives were tested for their neuroprotective action. A report showed that a nitro-HTyr was able to reduce ROS in an SH-SY5Y neuroblastoma cell line treated with H_2_O_2_ (100 μM, 30 min–3 h) and also was able to chelate several metal cations [259]. In another report, a series of alkyl nitrohydroxytyrosyl ether derivatives were synthesized from free HTyr and tested for their antioxidant effect in comparison with nitro-HTyr and free HTyr, using different assays, such as oxygen radical scavenging capacity (ORAC). It turned out that the antioxidant activity was dependent on the length of the alkyl side chain, as 2–4 carbon atoms showed the highest antioxidant activity, while those with 6–8 carbon atoms showed an efficiency lower than nitro-HTyr but higher than HTyr [260]. In a similar report, already mentioned above in the Section 6.2, pretreatment of rats with HTyr, HTyr acetate, or with nitroHTyr before intrastriatal infusion of MPP+ reduced oxidative stress, since dopaminergic stress symptoms decreased and the GSH/GSSG ratio increased; no clear difference between the effects of HTyr and derivatives was evident [149]. In another report, the antioxidant effects of five alkyl HTyr ethers (ethyl, butyl, hexyl, octyl, and dodecyl) were tested in rat brain slices against lipid peroxidation induced by ferrous salts and on reduction of GSH induced by diethylmaleate (DEM). It turned out that the alkyl HTyr ethers were 10-fold more antioxidant than HTyr, in particular the butyl derivative; the same antioxidant effects were observed after hypoxia-reoxygenation of brain slices [261]. A study even tested the possibility to render lipophilic HTyr in order to protect biomembranes from oxidation, since HTyr, being hydrophilic, is scarcely soluble in lipids. Two HTyr ester moieties were separated by a lipophilic alkyl spacer of different lengths, and the bis-ester derivatives resulted all able to protect liposome from induced oxidation, thus showing the potential to stabilize biomembranes in a free radical environment [262]. See Table 3.

## 11. HTyr and OLE in Senescence and Lifespan

### 11.1. Effects of HTyr and OLE on Senescence

Cellular senescence is characterized by a permanent arrest of cell proliferation due to stress conditions, including telomere shortening, oxidative stress, hypoxia, oncogenic activation, and DNA damage, and has been linked to processes such as tumor suppression, tissue repair, embryogenesis, and aging. The principal hallmarks of senescence are an increase in senescence-associated βgalactosidase activity (SA-β Gal), the presence of telomere dysfunction-induced foci (TIF), upregulation of specific cell cycle regulators (mainly the p53-p21 and p16-pRb axes), altered gene expression patterns, and the activation of a senescence-associated secretory phenotype (SASP), implying the synthesis and secretion of inflammatory mediators, growth factors, and extracellular matrix proteins [270,271]. The SASP is thought to play a major role in many age-related diseases, such as AD [272] and cancer [273], where it contributes to the maintenance of chronic inflammation, and in cardiovascular diseases and type 2 diabetes, also characterized by a low-grade chronic inflammation state.

It is worth noting that the elimination of senescent cells in progeroid and aged mice, through senolytic drugs or genetic mutations, markedly improves health span after senescent cells are cleared [270,274].

Several papers have reported that HTyr and OLE are endowed with the ability to modulate senescence or related inflammation in primary mammalian cell cultures, which represent suitable cellular models to study in vitro the mechanisms of aging. In pre-senescent human lung cells (MRC5) and neonatal human dermal fibroblasts (NHDFs), chronic (4–6 weeks) treatment with 1 μM HTyr or 10 μM OLE aglycone reduced some known markers of senescence, such as p16 and SA-β Gal. Moreover, the treatment decreased the secretion of senescence/inflammation markers such as IL-6 and metalloprotease, and levels of COX-2 and α-smooth-actin. Furthermore, the induction of inflammation following exposure to TNFα was abolished in OLE- and HTyr-pre-treated NHDFs [275]. Thus, the modulation of the senescence and senescence-associated inflammatory phenotype might be an important mechanism underlying the beneficial effects of olive oil phenols. Moreover, OLE (5 μM) and HTyr (1 μM) exerted a protective effect on 8 Gy irradiation-induced senescence, in γ-irradiated NHDFs, mitigating DNA damage and reducing the SASP mediated by the cyclic GMP AMP synthase/stimulator of intereferone genes (cGAS/STING)/NFκB pathway [276]. The ability of OLE and HTyr to alleviate senescence status, and the related SASP in normal cells can be exploited to improve the efficacy and safety of cancer radiotherapy.

Similar results were obtained in human dermal fibroblasts (HDFs) exposed to UVA. In this cell line, HTyr decreased SA-β-galactosidase activity and expression levels of MMP-1 and MMP-3 in a dose-dependent manner. These observations were accompanied by an anti-inflammatory effect, demonstrated by the reduced expression of IL-1β, IL-6, and IL-8 genes [277]. Interestingly, Varela-Eirín et al. (2020) demonstrated that OLE has senolytic activity in osteoarthritic chondrocytes (OACs) [278]. In fact, OLE reduced the inflammatory and catabolic factors mediated by NF-kB (IL-1ß, IL-6, COX-2, and MMP-3) and lowered cellular senescence in OACs, synovial and bone cells.

### 11.2. Effects of HTyr and OLE on Lifespan

Proliferative cellular lifespan is a complex process that is governed by multiple pathways in part interdependent with cellular senescence. In fact, normal somatic cells in vitro, after a finite number of cell divisions, can enter in the non-proliferative state of senescence [279].

In in vitro models, it was observed that treatment with OLE extended the lifespan of normal human fibroblasts (NHFs) by enhancing the activity of proteasome [280], which is known to decline during aging in human tissues as well as in senescent primary culture cells. Moreover, HTyr treatment was able to increase chronological lifespan (CLS) in quiescent NHFs, which have lost their capacity to repopulate. CLS measures the length of time nondividing cells survive before reach senescence, without telomere attrition. This process could be related to the observation that quiescent human somatic cells exhibit age-dependent loss in their regenerative capacity. The observed HTyr-induced extension of CLS was associated with an increase in manganese superoxide dismutase (SOD2) activity and reduction of age-associated mitochondrial ROS generation [281].

It is noteworthy that, as mentioned in previous sections of this review, HTyr and OLE have shown the ability to activate AMPK and SIRT1 pathways and inhibit the mTOR and insulin/IGF1 signaling, which are crucially involved in regulation of health span and, as a consequence, lifespan and senescence in animal models. Interestingly, numerous studies have been focused on the effects of HTyr and OLE on the lifespan of *Caenorhabditis elegans*. This free-living nematode is a powerful animal model mostly used in aging research due to its short life cycle. It was recently observed that OLE could significantly prolong the lifespan of *C. elegans* increasing the survival rates of worms against lethal heat shock and oxidative stress. OLE supplementation increased the expression and activity of antioxidant enzymes and suppressed the generation of malondialdehyde in nematodes. These positive effects on longevity and stress resistance were mediated by the factors DAF-16/FoxO, which plays a vital role in the insulin/IGF-1 signalling (IIS) pathway, and the SKN-1/Nrf2 pathway, another stress response and longevity pathway [282].

HTyr was also able to increase the survival of worms after heat stress and could further prolong the lifespan in unstressed conditions. In addition, in aged worms, exposure to HTyr and OLE led to the improvement of locomotive behavior and the attenuation of autofluorescence as a marker of ageing [154].

Moreover, numerous studies have been conducted on the beneficial effects of olive oil polyphenols in *C. elegans* models of PD and AD. Although *C. elegans* is not able to develop PD, exposure to the pesticide rotenone or the transgenic expression of human α-synuclein induces the Parkinsonian-like syndrome in *C. elegans*, which manifests as impaired movement. In models of PD, HTyr and OLE, as already mentioned in Section 6.2, have been shown to enhance swim performance of worms, to reduce α-synuclein accumulation in muscle cells, and to prevent neurodegeneration in α-synuclein-containing dopaminergic neurons [154]. Analogously, the treatment with a natural extract enriched in HTyr showed beneficial effects on the lifespan of wild-type nematodes and in a PD-model [155]. More recently, it was observed that in *C. elegans*, treatment with an olive fruit extract 20% rich in HTyr had beneficial effects on AD features such as Aβ- and tau-induced toxicity, as well as on oxidative stress. The treatment prevented oxidative stress and delayed Aβ-induced paralysis reducing Aβ aggregates. Indeed, the extract partially reduced the proteotoxicity associated with aggregation of the tau protein. The observed effects were correlated to the activity of the SKN-1/NRF2 transcription factor and the overexpression of HSP-16 [283].

## 12. Protective Effects of HTyr and OLE against Skeletal Muscle Dysfunction

In addition to its fundamental function in locomotion and maintenance of posture, skeletal muscle plays crucial roles in energy and glucose metabolism. This tissue has also emerged as an endocrine organ producing and releasing growth factors and cytokines (commonly termed myokines) that modulate systemic physiology. Furthermore, skeletal muscle may cross-talk with the nervous system and other tissues via direct muscle–nerve interactions, release of metabolites, systemic adaptation deriving from the energy demand of contracting muscle (e.g., during exercise), and myokines [21,284]. The age-associated loss of muscle mass and strength (called sarcopenia) is an important medical condition that results in fragility and reduced mobility and negatively affects bone mass, which also declines during aging causing osteopenia and osteoporosis. The concomitance of sarcopenia and osteoporosis in elderly patients leads to severe physical disability, frailty, high risk of falls, and osteoporotic fractures [285]. Furthermore, sarcopenia also influences global metabolic homeostasis, lifespan, systemic aging, and progression of age-related degenerative diseases in non-muscle tissue, as well as the organism’s responses to oxidative stress and dietary restriction [286].

### 12.1. Age-Related Sarcopenia and Osteoporosis

Multiple factors contribute to sarcopenia of aging, including chronic activation of inflammatory pathways, enhanced ROS generation and/or dysregulation of redox signaling, loss of muscle proteins resulting from an imbalance between protein synthesis and degradation, dysregulation of muscle autophagy, and impairment of neuromuscular transmission. These age-associated dysfunctions lead to myofiber death, muscle atrophy, and reduced regenerative potential [287,288].

A few studies have shown beneficial effects exerted on sarcopenia by olive mill wastewaters- and olive leaf-derived extracts rich in HTyr and/or OLE.

Pierno and colleagues (2014) tested the therapeutic potential of a polyphenolic mixture (LACHI MIX HT), extracted from olive mill wastewaters and containing mainly HTyr and low amounts of other phenolic compounds (tyrosol, catechol, gallic acid, homovanillic acid, and caffeic acid) in the amelioration of skeletal muscle dysfunctions due to aging-associated oxidative stress [289]. The effects of treatment of old rats with LACHI MIX HT were compared with those of purified HTyr. LACHI MIX HT treatment was able to counteract some of the alterations in excitation/contraction function typically observed in aged skeletal muscle. In particular, the treatment was shown to increase muscle weight, to improve sarcolemma chloride channel conductance, contractile function, and ATP-dependent potassium channel activity and to decrease blood creatine kinase levels and the resting cytosolic calcium concentration. Furthermore, LACHI MIX was more effective than purified HTyr in ameliorating most of the examined parameters, possibly due to the synergic activity of its various components.

Extracts from olive tree (*Olea europaea*) leaves are rich in OLE and HTyr, have a high antioxidant and scavenging power, and have been reported to exert beneficial effects on several health conditions, including hypertension, cardiovascular diseases, diabetes, and hyperlipidemia [290]. A recent study analyzed the effects of treatment with an olive leaf extract on aging-related sarcopenia and skeletal muscle insulin resistance [291]. Sarcopenia is known to be associated with: (i) increased expression of the E3 ubiquitin ligases Muscle Ring-finger protein-1 (MURF1) and Atrogin-1, which play a major role in the process of age-induced muscle loss [292]; (ii) increased expression of myostatin, a secreted myokine that inhibits skeletal muscle growth and negatively impacts muscle metabolism [293,294]; (iii) upregulation of histone deacetylase-4 (HDAC4), a deacetylase that promotes muscle atrophy in response to multiple stimuli by targeting several substrates, including myosin heavy chains (MyHC) and the transcriptional regulator PGC1α [295]; (iv) upregulation of the myogenic regulatory factors MyoD and myogenin, possibly as a compensatory mechanism [296]. A 21-day treatment of old rats with an olive leaf extract was shown to attenuate the age-induced atrophy of gastrocnemius muscle and the alterations in the mRNA expression of many of the above- mentioned markers of sarcopenia. Specifically, the authors observed a reduction of HDAC4, MyoD, myogenin, myostatin levels, and upregulation of PGC1α levels. The treatment also prevented the age-induced elevation of several pro-inflammatory markers and enhanced the expression of the anti-inflammatory cytokine IL-10 that was reduced in old untreated animals. Furthermore, the treatment improved muscle insulin sensitivity in gastrocnemius muscle by activating the insulin-dependent PI3K/Akt pathway [291].

In a follow-up study, the same authors evaluated whether the addition of the olive leaf extract to an oil mixture, composed of 25% algae oil and 75% EVOO (AO:EVOO), could synergistically attenuate sarcopenia in old rats [297]. In previous studies, the AO:EVOO mixture per se was reported to exert protective effects against cardiometabolic and skeletal muscle alterations associated with aging [298,299]. Combined treatment of 24-month-old rats with the olive leaf extract and AO:EVOO mixture provided further benefit to sarcopenia preventing not only the decrease in the weight of gastrocnemius muscle but also in the weight of the soleus muscle. Besides preventing the induction of the sarcopenia-related markers HDAC4, MyoD, myogenin, and rescuing the downregulation of PGC1α, the co-administration of both ingredients attenuated the reduction of MyHC isoforms, downregulated the expression of Atrogin-1 and MuRF1, prevented age-induced macrophage infiltration, and activated the insulin-dependent PI3K/Akt pathway in gastrocnemius muscle and adipose tissue [297].

There is ample evidence for a constant cross-talking between skeletal muscle and bone via paracrine and endocrine signals and for the existence of common pathogenic pathways shared by sarcopenia and osteoporosis including oxidative stress, increased inflammatory cytokine production, reduced anabolic hormone secretion, and reduced physical activity [300,301]. Consumption of olives, olive oil and olive polyphenols has been shown to improve bone health [302,303]. In particular, a number of studies reported physiological effects exerted by HTyr and/or OLE against osteoporosis. Hagiwara et al. (2001) evaluated the effects of HTyr and OLE on bone formation using cultured osteoblasts and osteoclasts, and on bone loss in ovariectomized mice, a widely used experimental model of post-menopausal osteoporosis [304]. They showed that HTyr and OLE inhibited the formation of multinucleated osteoclasts in culture, and that OLE enhanced the deposition of calcium by osteoblasts. Furthermore, OLE and HTyr were able to decrease bone loss in trabecular bone of femurs in ovariectomized mice. Other studies reported that OLE, HTyr, and olive mill wastewater extracts elicited beneficial effects on femur bone mineral density and improved inflammatory and oxidative stress markers in an experimental model of senile osteoporosis induced by ovariectomy associated with an acute inflammation in rats [305,306]. More recently, a beneficial effect of OLE on bone mineral density of lumbar vertebra and left femur of ovariectomized rats was also reported by Liu et al. (2022) [307]. Moreover, OLE could reduce the serum levels of IL-6 and malondialdeyde (a marker of lipid oxidation). In cellular experiments, it could also promote the proliferation of primary osteoblasts while inhibiting osteoclast differentiation by upregulating expression of osteoclastogenesis inhibitory factor (OPG) and downregulating expression of receptor activator for nuclear factor-*κ*B ligand (RANKL). Finally, a study conducted in the murine osteoblast MC3T3-E1 cell line revealed a cytoprotective effect of HTyr against oxidative stress-induced osteoblast apoptosis and shed light on the underlying mechanism [308]. In fact, HTyr was shown to prevent oxidative stress-induced mitochondrial dysfunction and osteoblast apoptosis by decreasing the cleavage of optic atrophy 1 (OPA1), one of the most important mitochondrial dynamics proteins, and by activating the Akt/glycogen synthase kinase 3 β (GSK3β) signaling pathway.

### 12.2. Skeletal Muscle Atrophy and Oxidative Stress 

In addition to aging, skeletal muscle atrophy is the consequence of various conditions, such as muscle disuse, strenuous exercise, denervation, neurodegenerative disease, muscular dystrophies, obesity, and diabetes.

Periods of immobilization are often associated with aging. Prolonged periods of muscle inactivity result in oxidative stress and chronic elevation of ROS production, primarily derived from mitochondria, within inactive muscle fibers. Such disturbed redox signaling critically contributes to disuse-induced muscle atrophy through several mechanisms, including (i) inhibition of muscle protein synthesis via repression of the anabolic Akt/mTOR signaling; (ii) activation of the Ca^2+^-activated proteases calpains; (iii) acceleration of protein degradation via the ubiquitin–proteasome system; (iv) activation of apoptosis; (v) stimulation of autophagy [309]. By using a rat model of unloading-induced muscle atrophy (3-week tail suspension), Liu et al. (2014) showed that 7 days’ reloading efficiently rescued skeletal muscle atrophy and mitochondrial dysfunctions [310]. Interestingly, administration of a mixture of mitochondrial nutrients including HTyr, α-lipoic acid, acetyl-L-carnitine, and coenzyme Q_10_ for 4 weeks to unloaded rats exerted a reloading-like effect and promoted the recovery initiated by reloading. Specifically, treated animals showed enhanced motor function, increased soleus muscle weight, decreased protein degradation, and apoptosis. Furthermore, the nutrient mixture rescued unloading-induced mitochondrial defects, by increasing total anti-oxidative capability, mitochondrial copy number and electron transport chain complex I and II activities and markers of mitochondrial biogenesis, such as PGC1α, nuclear respiratory factor 1 (NRF1), and mitochondrial transcription factor A (Tfam).

Redox-sensitive signaling pathways also play a pivotal role in exercise-mediated remodeling of skeletal muscle. Skeletal muscle shows different responses to the type of exercise, as well as its frequency, intensity, and duration. Increased ROS production in contracting muscle fibers plays a required role in skeletal muscle adaptations induced by regular endurance or resistance exercise training, while excessive ROS produced by strenuous or acute exercise can cause muscle oxidative stress, fatigue, and damage [309,311,312]. As reported by Feng et al. (2011), an 8-week intensive exercise program in rats resulted in decreased endurance capacity and highly induced expression of muscle atrophy (MURF1 and Atrogin-1) and autophagy markers (Atg7, Beclin-1 and LC3) [313]. This was accompanied by increased mitochondrial fission induced by excess of ROS in skeletal muscle and a decrease in PGC1α and complex I subunit expression. All of these changes were eliminated by treatment of exercised rats with HTyr. In addition, HTyr enhanced mitochondrial fusion and mitochondrial complex I and II activities and inhibited the expression of the oxidative-stress-responsive proteins p53, p21, and SOD2 in muscles of exercised rats [313]. Therefore, HTyr appears to improve exercise capacity by protecting contracting muscles from excessive ROS production.

### 12.3. Oxidative Damage and Skeletal Muscle Cell Degeneration

Protective effects of HTyr and HTyr derivatives on oxidative stress- and inflammation-induced mitochondrial dysfunction and muscle cell degeneration were also described in the C2C12 myogenic cell line.

Wang et al. (2014) reported that induction of oxidative stress in C2C12-differentiated myotubes promoted significant mitochondrial dysfunction in a time-dependent manner, accompanied by decreased expression of MyHC and myogenic regulatory factors (MyoD, myogenin, MRF4) and induction of apoptosis [314]. Furthermore, oxidative stress was shown to promote rapid cleavage of OPA1, a protein that controls mitochondrial inner membrane fusion and remodeling. Pre-treatment with HTyr acetate, which is present in olive oil at a concentration similar to that of HTyr, significantly prevented oxidative stress-induced OPA1 cleavage and mitochondrial morphology changes, and this was accompanied by improvement of mitochondrial oxygen consumption capacity, ATP productive potential and activities of mitochondrial complexes I, II, and V, and inhibition of oxidative stress-induced MyHC decrease [314].

Another study showed that pre-treatment with HTyr or the corresponding ester HTyr laurate effectively protected C2C12 myoblasts from apoptotic cell death induced by H_2_O_2_ treatment [105].

Furthermore, pre-treatment with HTyr was shown to counteract muscle cell degeneration induced by the inflammatory cytokine TNFα in C2C12 differentiating myoblasts by increasing the expression of differentiation markers (MyHC and myogenin), PGC1α and mitochondrial complexes I and II, as well as the activity of muscle creatine kinase [315].

Finally, a couple of studies investigated the effects of OLE in primary avian skeletal muscle cells. Kikusato and colleagues (2016) reported that treatment of avian muscle cells with OLE induced the mRNA expression of avian-specific uncoupling protein (avUCP), PGC1α, and downstream mitochondrial biogenesis-related genes (NRF1, Tfam, ATP5a1), as well as the activity of mitochondrial cytochrome c oxidase (COX) [316]. Additionally, the expression of the SIRT1 deacetylase was found upregulated by OLE. Since it is well-known that deacetylation by SIRT1 induces the PGC1α co-transcriptional activity [181], these results suggested that OLE upregulates mitochondrial biogenesis and avUCP expression in muscle avian cells through the SIRT1/PGC1α regulatory axis. Furthermore, OLE was observed to suppress the levels of mitochondrial ROS generation, possibly via the up-regulation of SOD2 and avUCP gene expression. In a follow-up study, Muroi et al. (2022) used a specific inhibitor of SIRT1 activity to confirm that indeed it can mediate the enhancement of PGC1α, avUCP, ATP5a1, and SOD2 gene expression and the suppression of mitochondrial ROS generation induced by OLE in chicken muscle cells [317]. The study also showed that the action of OLE on the above markers of mitochondrial biogenesis and oxidative damage can be mediated by transient receptor potential cation channel subfamily V member 1 (TRPV1), as determined by using an antagonist of TRPV1. Furthermore, treatment of chicken muscle cells with OLE was shown to increase intracellular Ca^2+^ concentration and stimulate the phosphorylation, and hence activation, of the AMPK kinase in a TRPV1-dependent manner. Based on the knowledge that Ca^2+^/calmodulin-dependent protein kinase 2 induces phosphorylation of AMPK and that AMPK, SIRT1, and PGC1α are part of a highly coordinated regulatory network [181,318], the authors proposed that the effects of OLE in chicken muscle cells may be due to induction of Ca^2+^ influx, possibly through the activation of TRPV1 localized at ER, and increased activity of AMPK, followed by SIRT1 activation and then activation of PGC1α and induction of downstream mitochondrial targets [317].

### 12.4. Skeletal Muscle Insulin Resistance and Metabolic Syndrome

As mentioned earlier, skeletal muscle plays a crucial role in whole-body energy homeostasis and is a primary tissue of insulin-induced glucose uptake and oxidative metabolism. Muscle atrophy and sarcopenia are associated with the development of skeletal muscle insulin resistance [319], which is considered to be a major driver of metabolic syndrome (MetS), a cluster of medical conditions (obesity, hypertension, dyslipidemia, hyperglycemia) that together increase the risk of developing type 2 diabetes, cardiovascular diseases, and stroke. Major factors in the development of MetS include sarcopenia, chronic inflammation, abdominal obesity, insulin resistance/hyperinsulinemia, physical inactivity, high fat intake, and genetic factors. MetS, obesity, and diabetes induce adverse effects on skeletal muscle function, including muscle fiber atrophy and contractile dysfunctions, altered metabolism, insulin resistance, oxidative stress, mitochondrial dysfunctions, and reduced regenerative potential [319,320,321,322,323].

Many studies have reported beneficial effects exerted by HTyr and OLE in various cellular and animal models of MetS and diabetes [324,325,326,327]. Here, we focus on a few studies where the effects of HTyr and OLE on insulin resistance and MetS were examined specifically in skeletal muscle tissue or skeletal muscle cells.

Commonly used MetS murine models exhibit metabolic disorders induced by high-fat diet (HFD) feeding or a non-functional leptin pathway, such as the *db*/*db* mouse [328]. Cao and colleagues extensively examined the effects of HTyr in skeletal muscle and liver of C57BL/6 mice fed a HFD with or without supplementation of low-dose or high-dose HTyr (10 or 50 mg/Kg/day, respectively) for 17 weeks [329]. HTyr administration was shown to effectively inhibit body and organ weight increase (50 mg/Kg/day). Both low- and high-dose HTyr treatment enhanced glucose tolerance and lowered the serum levels of glucose, insulin, lipids, and inflammatory cytokines. In skeletal muscle and liver, HTyr reduced the accumulation of lipid deposits through inhibition of the sterol regulatory element-binding transcription factor (SREBP) pathway, attenuated oxidative stress by enhancing antioxidant enzyme activity, elevated the decreased expression of complex I and II subunits of the electron transport chain induced by HFD, and lowered the expression of markers of mitochondrial fission and apoptosis. Moreover, in muscle tissue, HTyr decreased the level of mitochondrial protein carbonylation and elevated the activities of complexes I, II, and IV. The effects of HTyr were also examined in diabetic *db*/*db* mice treated for 8 weeks with low-dose HTyr (10 mg/Kg/day) or with metformin, a commercial antidiabetic drug, as a positive control. Both HTyr and metformin reduced the fasting blood glucose levels in *db*/*db* mice, whereas the fasting serum levels of lipids (triglyceride, cholesterol) were lowered by HTyr but not by metformin. Furthermore, HTyr was shown to be more effective than metformin in reducing protein and lipid damage in skeletal muscle and liver and in increasing the activities of mitochondrial complexes I and IV [329].

Another study evaluated the effects of HTyr acetate (HTyr-Ac) on glucose consumption in C2C12 skeletal muscle cells and 3T3-L1 adipocytes. Treatment with increasing concentration of HTyr-Ac was reported to stimulate glucose uptake in a dose-dependent manner in both C2C12 myotubes and differentiated adipocytes. Furthermore, HTyr-Ac was shown to exert an anti-adipogenic effect by inhibiting adipocyte differentiation and by stimulating lipolysis of fully differentiated adipocytes [330].

With regard to OLE, it has been shown that C57BL/6 mice fed an HFD supplemented with 0.038% of OLE, for 12 weeks, exhibited reduced levels of fasting glucose and improved insulin resistance. Moreover, the gastrocnemius muscle of OLE-fed mice displayed increased expression and membrane localization of glucose transporter type 4 (GLUT4) protein [331]. In the same study, C2C12 myotube cells treated with OLE displayed enhanced glucose uptake and GLUT4 translocation to the plasma membrane. Furthermore, OLE improved insulin sensitivity in C2C12 cells treated with palmitic acid, a model of lipotoxicity and insulin resistance. OLE did not show a synergistic effect with insulin regarding glucose uptake, and stimulated phosphorylation, and hence activation, of AMPK but not phosphorylation of the Akt kinase, which is downstream of the insulin signal [331]. Therefore, the mechanism through which OLE can induce glucose uptake in muscle cells appears to be linked to the activation of AMPK but does not require activation of the PI3K/Akt pathway.

Similar results were reported by Hadrich et al. (2016) who showed that treatment of C2C12 myotubes with OLE can stimulate glucose uptake and activate AMPK and MAPK signaling but not the PI3K/Akt insulin signaling pathway [332]. In addition, the treatment was shown to protect C2C12 myotubes against oxidative stress induced by H_2_O_2_ by decreasing ROS production and lipid peroxidation levels.

In agreement with the previously discussed studies, Alkhateeb et al. (2022) have recently reported protective effects of OLE on muscular insulin resistance in isolated soleus muscle preparations treated for 12 h with a high concentration of palmitate [333]. In fact, OLE treatment enhanced insulin-stimulated glucose uptake, the translocation of GLUT4 at the plasma membrane, and the levels of Akt substrate of 160 kDa (AS160) phosphorylation. Furthermore, OLE promoted the activation of AMPK and an inhibitor of AMPK blocked OLE-stimulated glucose uptake, GLUT4 translocation, and AS160 phosphorylation. This indicated that OLE can ameliorate palmitate-induced insulin resistance via an AMPK-dependent mechanism.

As mentioned earlier, OLE is the most abundant phenolic compound in extracts from olive leaves. By using a rat model of STZ-induced diabetes, Giacometti et al. (2020) investigated the effects of an OLE-rich olive leaf extract on GLUT4 expression and intracellular vesicular GLUT4 trafficking in soleus muscle [334]. Diabetic rats treated for 10 days with 512, 768 or 1024 mg/Kg of olive leaf extract (containing 20.3, 33 or 44.5 mg/kg of OLE, respectively) displayed significantly reduced blood levels of glucose and triglycerides. Histopathological examination of the diabetic soleus muscle revealed a decrease in fiber size and increased fibrosis, and such pathological muscle alterations were improved after treatment with the olive leaf extract in a dose-dependent manner. Furthermore, treatment with olive leaf extract was shown to promote GLUT4 translocation to the myofiber membrane in soleus muscle and to enhance the expression levels, and the colocalization with GLUT4, of the Rab GPTases Rab8A, Rab13, and Rab14, which are central regulators of vesicular transport along exocytic, endocytic, and recycling pathways [334,335] 

The effects of HTyr, OLE, and nutrient mixtures reported in the studies discussed above are summarized in Table 4.

## 13. Effect of HTyr and OLE in Gut Microbiota–Brain Axis

### 13.1. Gut Microbiota

Humans are defined as super-organisms or holobionts that live in harmony with their symbiotic roommates represented by more than 100 trillion microorganisms whose coordinated actions are thought to be important to human life [336]. Altogether they form anatomical, physiological, immunological, or evolutionary units. In particular, the human gut harbors a bacterial ecosystem of 10^13^–10^14^ bacterial cells and it is also populated by viruses, fungi, and protozoa that collectively form a complex microbial community known as the gut microbiota [337]. All these microorganisms are not identified as pathogens by our immune system but, on the contrary, most of them coexist symbiotically with the enterocytes [338].

The presence of an enormous number of microbes, which far exceeds the number of human cells, leads to the assumption that the microbiota can influence the physiology of the host organism. This is even more reasonable if we consider that the microbial genome (microbiome) vastly exceeds the human host genome’s size [339]. This element alone is sufficient to realize that our symbiotic bacteria are essential for numerous physiological processes that guarantee well-being by ensuring our body’s homeostasis. An imbalanced composition of bacterial populations can be harmful to human health, contributing to the onset of numerous pathologies. However, although most alterations of the gut microbiota turn into an outbreak of diseases and pathologies, actually, a modification in the composition of the gut microbiota is a physiological event. It has been reported that the gut microbiota plays critical roles in the maintenance of human health: (i) taking part in the digestion of food substances, facilitating access to nutrients that would otherwise be inaccessible to the host; (ii) promoting host cells differentiation to protect them from pathogens; (iii) stimulating and modulating the immune system [340].

The human intestine is composed of a balanced microbiota with two dominant phyla accounting for about 90% of the total, i.e., Bacteroidetes and Firmicutes, and four less represented phyla, such as Proteobacteria, Actinobacteria, Fusobacteria, and Verrucomicrobia [341]. Such microorganisms can be autochthonous (indigenous) or allochthonous (transient) and are symbionts in most cases. Despite this, they are considered pathogenic when assuming opportunistic behavior to the detriment of the host [337]. The Firmicutes/Bacteroidetes ratio is an important parameter for emphasizing a potential gut microbiota disorder [342,343,344,345] but the abundance, diversity, and homogeneity of the intestinal microbiota are also indicators of a state of health. Nevertheless, this ratio can also be linked to physiological changes in bacterial profiles during different stages of life. Indeed, it is low in the first years of life, then increases in adulthood and decreases again in old age [346]. Overall, the composition of the microbiota is unique to each individual because it can be influenced by different factors acting throughout life [347]. Indeed, there are many factors such as genetics, diet, environment, exposure to drugs or more generally lifestyle, which influence the composition of the microbiota allowing the proliferation of certain species, rather than others. This aspect is extremely important as the conditions that favor the balanced assembly of microbial populations beneficial to the host determine the establishment of homeostasis defined as “eubiosis”. If, on the other hand, factors come into play that destabilize this condition of equilibrium, the so-called state of “dysbiosis” occurs. Changing the composition and function of the gut microbiota can alter intestinal permeability, digestion, and metabolism as well as immune responses. In particular, an altered state of the gut microbiota causes a pro-inflammatory state, and this condition can lead to the onset of many diseases ranging from gastrointestinal and metabolic conditions to immunological and neuropsychiatric diseases [348].

### 13.2. Gut Microbiota across the Lifespan

The intestinal microbiota establishes a co-evolution relationship with the host (which in fact represents its ecosystem) and its development is regulated by a complex interaction between the host and environmental factors, such as diet and lifestyle. For this reason, knowing the transformation of the gut microbiota from birth to old age may shed light on the variation of this community during lifespan and on the possible associations with disease risks. The symbiosis with the microbiota is established from birth and is rewired several times in the first years of life, a period during which children undergo rapid and irreversible growth, showing significant increases in height and weight, and their organs and cognitive abilities undergo great changes [349]. Numerous scientific papers reported that the establishment of this symbiosis in childhood is of the utmost importance and that imbalances in intestinal microbiota composition during infancy are associated with various metabolic, immune, and neurological diseases. Thus, early childhood offers a unique opportunity to modulate the gut microbiota in order to promote long-term health [349]. At first, the intestinal microbial flora of the newborn shows a low biodiversity, which however will increase during development. During the first days of life, Proteobacteria and Firmicutes represent the two most abundant phyla in vaginally born infants; while from the 7th to the 15th day after birth, Actinobacteria appeared in the feces of cesarean-delivered infants [350]. Around the age of three, the intestinal microbiota will move from a highly unstable and poorly differentiated composition to a more stable composition with the typical characteristics of the adult intestinal microbiota [351]. From this point on, the gut microbiota will rest in a stable state from the third to the seventh decade of life, although the proportions of *Bifidobacteria*, *Firmicutes*, and *Fecalibacterium prausnitzii* tend to decrease with an increase in *Escherichia coli*, *Proteobacteria,* and *Staphylococcus* [338,352]. Research carried out on the dynamics of the microbiota in elderly individuals has also provided further information on the possible trajectories of the intestinal microflora throughout human life [353]. While the composition of the adult human gut microbiota is generally stable if unperturbed, its stability deteriorates in old age [352] and alterations causing dysbiosis are becoming more and more frequent due to age-related factors. For example, a decrease in *Bifidobacterium* and an increase in *Clostridium* and *Proteobacteria* have been observed in older people [338]. Given the role of *Bifidobacterium* in stimulating the immune system and metabolic processes, its decrease could partly explain the compromised immune system in the elderly [341]. Furthermore, the greater possibility of developing dysbiosis in old age is a factor to be correlated to the onset of neurodegenerative diseases.

### 13.3. Gut Microbiota–Brain Axis

The trade-off between gut microbiota and brain is now regarded as a pivotal hub for healthy life and aging. The so-called microbiota–gut–brain axis is considered a neuroendocrine system and implies a bidirectional communication between the gut microbiota and the brain, whose dysregulation has emerged to affect host health and disease [354]. A healthy host–microbial balance is fundamental to maintain the physical and mental health of both young and elderly populations, while dysbiosis is steadily more implicated in the onset of metabolic, inflammatory, and neurological disorders [347]. It is now clear that alterations in top-down (brain to gut) communications are associated with gut inflammation syndromes and appetite disturbances, whereas dysregulations in the bottom-up (gut to brain) interactions are associated with alterations in nervous system functions and neurologic pathologies [355]. The relationship between microbiota and brain is known to be regulated at multiple levels, such as immunological (cytokines), endocrine (cortisol), and neuronal, including both central (CNS) and enteric (ENS) nervous systems interconnected by the vagus nerve [354].

The short-chain fatty acids (SCFAs), formed by microbial processing of dietary indigestible fibers, are emerging as key players in neuro-immunoendocrine regulation. Indeed, SCFAs can be used locally by colonocytes as an energy source but are also effective in the maintenance of host intestinal barrier integrity and immunity, suppressing cytokines production by myeloid cells and inducing regulatory T-cell differentiation [356]. At the colon level, SCFAs also induce the secretion of anorexigenic peptides, which act on hypothalamic centers regulating nutritional habits and energy balance [357]. In addition to local effects, SCFAs can enter the bloodstream to be distributed to other organs and, after crossing the blood–brain barrier (BBB), contribute to its integrity, inducing the expression of tight junction proteins, and can modulate brain and behavior [358]). Moreover, in the CNS, SCFAs influence neuroinflammation by regulating the maturation and function of microglia as well as by modulating the levels of neurotrophic factors, increasing neurogenesis, contributing to the biosynthesis of serotonin, and improving neuronal homeostasis and function [356].

The gut microbiota may influence the functions of ENS and CNS nervous systems also by producing metabolites and neurotransmitters with neuromodulatory properties, such as gamma-aminobutyric acid (GABA), noradrenaline, dopamine, serotonin (5-HT), and their precursors (e.g., tryptophan and tyrosine). In turn, the host nervous system modulates the motility of the gastrointestinal (GI) tract and the intestinal barrier homeostasis, sustaining the microbial community [355].

Studies usin g germ-free (GF) mice (i.e., axenic, free of all microorganisms) or antibiotic-treated specific pathogen-free (SPF) mice (i.e., free of a specific list of pathogens) have provided the strongest proof of the importance of microbiota in gut–brain signaling. GF mice have been demonstrated to have altered behavior, impaired immune systems, dysregulated hormone signaling, abnormal metabolism and neurotransmission with respect to their standard counterparts [359]. For example, GF mice were reported to show hyperactivity of the hypothalamic–pituitary–adrenal (HPA) axis upon restraint stress, which could be reversed by administration of *Bifidobacterium infantis*, but not by monocolonization with the enteropathogenic bacteria *E. coli* [360,361], highlighting the influence of microbiota on stress responsivity. In addition, a complete SPF flora was able to partly reverse the HPA response to stress only when it was introduced at an early stage of development, suggesting that brain sensitivity to gut signals may occur only within a critical time window [360]. Anxiety-like behaviors were also found to be affected in GF mice, which frequently showed an increased exploratory and locomotor behavior, as an index of reduced anxiety [362,363,364]. Furthermore, altered expression of synaptic plasticity-related genes, including BDNF and nerve growth factor-inducible clone A (NGFI-A) [364,365], as well as variable plasma levels of neurotransmitters [363], have been reported in GF mice. However, it is worth noting that the results of the studies on GF mice may vary depending on age, gender, and strains used, as reviewed by Cryan et al. (2019) [347].

Besides being an important risk factor for the development of many diseases, ranging from cardiovascular complications to neurologic illnesses, gut dysbiosis has been also demonstrated to affect post-disease recovery. For example, emerging experimental and clinical evidence showed the influence of gut microbiota not only on ischemic stroke pathogenesis but also on treatment outcomes. In fact, ischemic brain dysregulates intestinal homeostasis directing aberrated signals to the intestine either via the neural or HPA axis pathways, resulting in poor stroke treatment prognosis [366]. On the other hand, the transplantation of gut microbiota from normal mice into the intestinal tract of mice with ischemic stroke improved the long-term prognosis and survival rate [367]. Therefore, therapeutic approaches targeting gut dysbiosis can be considered as promising tools for the treatment and management of stroke or, more in general, of various age-related disorders in which gut dysbiosis may have a role, including neurodegenerative, cardiovascular, metabolic, and musculoskeletal diseases, as well as immune system diseases and cancer [368,369,370,371].

### 13.4. Effects of HTyr and OLE on Gut Microbiota under Pathological Conditions

It is now well-established that dietary habits represent one of the main factors determining the composition of the gut microbiota. The Mediterranean diet, which is rich in foods of plant origin, is universally recognized as healthy, and it is amply demonstrated that EVOO, its main source of fat, brings benefits to human health influencing the gut microbiota composition [372]. In general, ingested polyphenols have been reported to counteract oxidative stress and inflammatory injury, thus protecting the digestive system from chronic and recurrent diseases. Furthermore, it is widely recognized that they are able to exert neuroprotection and promote cognitive functions, but their connection with the gut microbiota is a more recent discovery. Several studies reported the beneficial effect of EVOO or, more in general of polyphenols, on intestinal microbiota [72,372,373,374], whereas less is known about the effect of the exclusive administration of HTyr and OLE. Thus, in this review, we will focus on the outcome of HTyr and OLE on gut microbiota shaping and its implications on healthy host aging. Unlike HTyr and tyrosol, which are well absorbed in the small intestine, OLE reaches the colon unchanged and is rapidly degraded by the colonic microflora to produce HTyr, which can then be absorbed [375]. OLE is preferentially degraded in vivo by lactic acid bacteria, such as *Lactobacillus* and *Bifidobacterium* species [376], which use OLE as a carbon source and benefit from its metabolism. HTyr administration in high-fat-diet-fed mice was found to increase the concentration of the probiotic bacteria *Lactobacillus*, specifically *L. johnsonii*, and to not significantly change the ratio Firmicutes/Bacteroidetes (F/B), which was instead decreased in normal-diet-fed mice, indicating that HTyr may be beneficial for metabolism and may control F/B ratio [377]. Dietary HTyr supplementation in high-fat-diet-fed mice also reduced the numbers of *Proteobacteria* and *Deferribacteres*, whose abundances have been associated with dysbiosis in hosts with metabolic or inflammatory disorders [378], and promoted intestinal integrity reducing inflammation level [377]. The ability of HTyr to stabilize the F/B ratio and restrain the increase of *Proteobacteria* may also help controlling gut dysbiosis in inflammatory bowel disease patients, in which a decrease of Firmicutes and Bacteroidetes and an increase of *Proteobacteria* are the most consistent hallmarks of disease [379]. The beneficial effect of HTyr and OLE on gut microbiota is also boosted by their ability to inhibit pathogenic bacteria growth, either Gram-positive or Gram-negative, such as *E. coli* [72,380].

The HTyr-induced modulation of colon microbiota was also correlated to its powerful antioxidant effect in oxidative-stressed mice [178]. In fact, the HTyr-promoted decrease of the lipid peroxidation marker malondialdehyde (MDA) in the serum was correlated with the increase of the relative abundance of Firmicutes and *Lactobacillus* and the decrease of Bacteroidetes, as an effect of HTyr administration [178]. Moreover, HTyr decreased *Parabacteroides*, whose abundance was associated with the progression of oxidative stress and inflammation [178]. Interestingly, the HTyr-induced increase of Firmicutes positively correlated with a higher concentration of butyrate, one of the most important SCFAs, involved not only in maintaining intestinal function but also in reducing oxidative stress [381]). Surprisingly, HTyr supplementation was demonstrated to be effective in relieving oxidative stress in mice also by increasing *Staphylococcales* [382], whose success as pathogens is due in part to their ability to mitigate endogenous and exogenous oxidative and nitrosative stress [383]. The restrain of oxidative stress due to HTyr administration was also found to prevent fine dust (PM_2_._5_)-induced adiposity and insulin resistance in adult mice [384]. Once again, it was highlighted that gut microbiota may mediate the actions of HTyr, which was effective in increasing microbiota richness, reducing pathogenic bacteria, and reversing PM_2_._5_-induced gut dysbiosis. In particular, the genus *Akkermansia*, belonging to the Verrucomicrobia phylum and known to be beneficial for reducing weight gain and endotoxemia in mice and humans [385], was exclusively abundant in HTyr-administered mice. In addition, *Ruminococcaceae* and *Mycoplasmataceae*, beneficial for glucose metabolism and abundant in long-living people, were enhanced by HTyr administration as well as *Prevotellaceae*, which negatively correlated with lipid peroxidation biomarkers, thus contributing to the HTyr antioxidant effect [384]. Besides HTyr, OLE was also found to ameliorate advanced-stage type 2 diabetes in mice by regulating gut microbiota [386]. Consistent with the previously reported effects of HTyr administration, O LE was shown to significantly reduce the relative abundance of Bacteroidetes, including the *Parabacteroides* genus, and increase that of Verrucomicrobia, including the *Akkermansia* genus, confirming their importance in improving diabetes symptoms and insulin resistance [386].

Altogether, the above studies highlighted that HTyr and OLE both act as prebiotics in gut microbiota modulation, favoring intestinal microbes’ homeostasis, even in the presence of chronic diseases. Their administration has been demonstrated to increase the diversity of gut microbiota, enhancing beneficial bacteria, and inhibiting pathogenic ones, thus impacting the health of multiple organs and systems. In fact, the well-established relationship between gut and CNS highlights the role of prebiotics in reducing the degree of oxidative stress and consequent progression of neurodegenerative and cardiovascular diseases and metabolic syndromes. For this reason, the maintenance of a condition of eubiosis is crucial to guarantee healthy aging, preventing and ameliorating several chronic pathologies. The potential therapeutic application of prebiotics supplied with the diet is increasingly investigated in order to keep optimal health during aging and also for the treatment of age-related disorders.

## 14. Conclusions

A large body of evidence has highlighted that HTyr and OLE, a low-molecular weight biologically active phenol and a secoiridoid found into EVOO and olive mill wastewater, can be considered potential candidates as anti-aging drugs given their demonstrated beneficial effects on numerous chronic age-associated pathologies. In particular, the literature analyzed in this review shows that HTyr and/or OLE can prevent or delay several age-related neurodegenerative and neurological diseases, skeletal muscle and bone deterioration, and altered metabolism, can increase lifespan in model organisms and antagonize cellular senescence, and can modulate the gut microbiota.

In the nervous and musculoskeletal systems, HTyr and OLE exert their beneficial effects by counteracting different features of aging, including chronic inflammation, increased oxidative stress and mitochondrial dysfunction, neural stem cell exhaustion, imbalanced protein synthesis and degradation, and decreased autophagy. Such pleiotropic actions of HTyr and OLE rely on their ability to activate crucial regulators of the main molecular pathways involved in autophagy, energy metabolism, mitochondrial biogenesis, antioxidant defense, such as SIRT1, AMPK, FOXO, PGC1α, and Nrf2.

The microbial communities of the gastrointestinal tract can influence the host physiology through their ability to produce and modify multiple metabolites and compounds, and an altered state of gut microbiota can contribute to the onset of degenerative disorders of the brain and other biological systems. Emerging evidence over the past decade indicates that HTyr and OLE can modulate the composition of gut microbiota. Further studies are required to clarify whether some of the beneficial effects of HTyr, OLE, or their metabolites are mediated by specific gut microbiota. In addition, it would be interesting to evaluate the potential application of HTyr, OLE and/or their derivatives as prebiotics to be provided in the diet to maintain good health during aging and to counteract age-related disorders.

## Data Availability

Not applicable.

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
