# Peer review of "Role of Hydroxytyrosol and Oleuropein in the Prevention of Aging and Related Disorders: Focus on Neurodegeneration, Skeletal Muscle Dysfunction and Gut Microbiota"

_nutrients, 2023, doi:10.3390/nu15071767_

Round 1

Reviewer 1 Report

The review article by et al. Micheli L. et al.” Role of hydroxytyrosol and oleuropein in the prevention of aging and related disorders: focus on neurodegeneration, skeletal and muscle dysfunction and gut microbiota” summarizes the results of studies that investigate the effects exerted by EVOO polyphenols, in particular hydroxytyrosol (HTyr) and oleuropein (Ole), on several age-associated neurodegenerative diseases. Although the topic appears timely and may  be of interest  to the readers of Nutrients, the revision process  has arisen several critical issues that need to be addressed by the authors in order to make this review article more focused and to increase its impact on the related field.

General Comments

Introduction

Lines102-123 Among aging-related diseases, Authors should also consider some pathological conditions associated with altered bone remodeling processes including osteoarthritis, osteoporosis and other disorders (see for instance Edwards et alBone 201580, 126–130; Guo et al. Sig. Transduct Target Ther 7,391 (2022); Chin et al. J Clin Med. 2022;11(21):6434). In particular, idiopathic osteoporosis of the elderly, which is a significant  risk factor for fragility fractures, is a  relevant global public health problem, being the economic burden of osteoporosis-related fracture enormous. (Clynes  et al,. Br  Bull. 2020;133(1):105-117). Although the current therapeutic options based on the use of anti-resorptive and/or anabolic agents, are effective in preventing bone loss, there is increasing concern about their long-term safety (see Skjødt  et al. e. Br J Clin Pharmacol. 2019 Jun;85(6):106). Hence, the need to find  new molecules endowed with  low  toxicity and limited side effects.  In this setting, growing experimental studies have shown that HTyr and OLE can target several signalling pathways involved in the modulation of bone remodelling processes in normal and pathological conditions including malignant bone diseases (see for instance  Casado-Díaz et al., Food Funct. 8 (2017) 1254; Castejón et al. , Antioxidants (Basel). 9 (2)  (2020) 149; Nicolin V et al. Front. Endocrinol. (Lausanne) 10 (2019) 494); Scoditti et al. Arch. Biochem. Biophys. 527 (2012) 81; Torre E. , Phytochem. Rev. 16 (2017), 1183–1226; M. Svobodova et al , Genes Nutr. 9 (2014) 376;  Xiong ZC Drug Des. Develop. and Ther. 13 (2019) 1879.). These findings suggest  a possible therapeutic role of these molecule in the prevention and  treatment  of various  bone disorders. In line with these observations recent preclinical and clinical studies have provided evidence on the beneficial effects exerted by HTyr and OLE in the prevention and treatment of  age-related bone disorders (see for instance Hagiwara et al. Eur. J. Pharmacol. 662 (2011) 78–84; Chin KY et al. , Int. J. Environ. Res. Public Health 13 (2016); Chin KY et al.,  Nutrients 9 (2017);  Nicolin et al. Front. Endocrinol. (Lausanne) 10 (2019) 494),   including cancer-related bone diseases (J.M. Morana, Nat. Prod. Commun. 11 (2016) 491–492; Leto et al.  Life Sci  2021, 1;264:118694; Przychodzen et al, Anticancer Res. 2019 39(3):1243-1251).

Authors should revise this section according to the new concept    emerged  following the revion of the paper.

2. Chemistry of hydroxytyrosol (HTyr) and oleuropein (OLE)

Lines 129- 130  several types of cancer, inflammation, cardiovascular and neurodegenerative diseases insert some reference related to these studies)

line 169  producing glucose,  Ole aglycone, HTyr and elenolic acid (Scheme 1)

3. Pharmacokinetics of HTyr and OLE

 3.1  Lines 200-229 .  The general aspects of  ADME should be omitted. It is presumed that these concepts are well known by the readers.

Line 236: resident time (?) : residence time

Line 245 and the arrives: better reaches

Line 268 Its capillary  (?) distribution: it would mean widespread distribution?

3.3. Metabolism and distribution

There are some important  aspects of Ole metabolism that need to be to highighted and   in particular :

1)   The potential therapeutic effectiveness of  this molecule is closely related to the possibility that it may reach in adequate concentrations its specific molecular targets in human tissues.  The probability of reaching its  key molecular targets in human tissues at a sufficient dose, is related to its metabolism and bioavailability. In fact, Ole and HTyr undergo extensive phase I and phase II metabolic processes that may affect their bioavailability and the systemic transfer at adequate concentrations to the target tissues, ultimately blunting their therapeutic effectiveness.  In order to overcome these hurdles, an increasing number of studies are currently  undertaken with the aim to develop new semi-synthetic derivatives endowed with abetter bioavailability, improved biological activity and, novel drug delivery systems based on  nanotechnology  (see Bonechi et al  Biophys. Chem. 246 (2019) 25D;  De Luca. Et al.Polymers (Basel). 2022 Apr 23;14(9):1726.  Karković Markovićet al. Molecules. 2019;24(10):2001.Monteiro  et al. .Antioxidants (Basel). 2021;10(3):444 Palagati et al. Daru J. Pharm. Sci. 27 (2019) 695 -708,  Russo et al. J. Pharm. Pharmaceutics 3(1) (2016) 40)

2)   Malliou et al. have recently shown that the a long-term  oral administration of Ole to  male 129/Sv WT mice  induced the synthesis of  the major cytochrome P450s (P450s)  in the liver via activation of PPAR α and other cellular factors, such as AHR, CAR, RXR, and PXR (Malliou et al. Drug Metab Dispos. 2021;49(9):833). This effect could modify the pharmacokinetic profile of co-administered drug whic are substrates of the P450s, thus altering their therapeutic efficacy and toxicity.  Therefore, these effects  shoud be taken into account and in particular  when Ole is co- administered  with drugs with  narrow therapeutic index

3.5. Toxicity

Authors  should discuss the implications of the metabolic aspects of  Ole and Htyr reported above in relation  to a possible future clinical use of these molecules in  differentage-related diseases. Beside, the possibility to take advantage of the various effects of Htyr and Ole on normal and malignant human cells  for therapeutic purposes should be also briefly mentionrd

Section 4 to Section 10

The effects exerted by  Ole  in some of  the  aging-associated  diseases  have not been  mentioned in detail.  An exaustive updated description of  the therapeutic potenial  of Ole in the treatment  of  acute and chronic neurodegenerative and neuropsychiatric disorders as well on mechanisms by which Ole may prevent neurodegeneration are provided by some recent excellent  review  articles ((see Angeloni et al. Int.  J.  Mol.  Sci, 18(11), 2230,2017  Butt et al.  J Food Biochem. 2021;45:e13967. See also Achour et al. Int. J. Mol. Sci. 2016;17:129; (Singh et al Sci Rep. 2023 Feb 11;13(1):2452); Giacometti and Grubić-Kezele Oxid Med Cell Longev. 2020 Jul 30;2020:6125638. Authors should integrate and discuss these  informations.

13. Effect of HTyr and OLE in gut microbiota-brain axis

Emerging experimental and clinical evidence indicates that gut dysbiosis  may have a role in the pathogenesis Ischemic Stroke ( see Chidambaram Cells. 2022 Apr 6;11(7):1239) Wang et al. Front Immunol. 2022 Mar 28;13:845243) as a restoration of the gut microbiome usually improves stroke treatment outcomes by regulating metabolic, immune, and inflammatory responses via the gut–brain axis (GBA). Ole has  been reported to be endowed with  antimicrobial activity against a wide number of  bacteria, (either gram+ or gram-) mycoplasma, and viruses  (Omar, S.H.. Sci. Pharm. 201078;133 Farràs M,. Nutrients. 2020 Jul 23;12(8):220). These findings imply that  Ole and Htyr  may be of potential clinical interest  for  the treatment of various  age-related disorders in which gut  dysbiosis may have a role including neurodegenerative diseases, cardiovascular diseases, metabolic diseases, musculoskeletal diseases, and immune system diseases and cancer (  see Chidambaram 2022; Varesi A, Int. J. Mol. Sci. 2022; 23(20):12289. Di Meo et al., Curr. Drug Metab. 19 (6), 2018;  Memmola. Nutrients. 2022 Sep 10;14(18):3749. )

14. Conclusions

Authors should revise this section according with the new  issue emerged  following the revion of the paper 

References

López-Otín C, Blasco MA, Partridge L, Serrano M, Kroemer G. The hallmarks of aging. Cell. 2013 Jun 6;153(6):1194-217. doi:

10.1016/j.cell.2013.05.039.

 Lines 52-55 and 2196  Ref 192 López-Otín C, Blasco MA, Partridge L, Serrano M, Kroemer G. The hallmarks of aging. Cell. 2013 Jun 6;153(6):1194-217. There is an updated version of this article  (López-Otín  C.  et al. Cell. 2023 Jan 19;186(2):243-278. doi: 10.1016/j.cell.2022).  These authors report that the  number of hallmark of aging should be updated to 12 ( as compared with the previous number, e.g.,9)

Minor point

Lines 435-437….  as also indicated by the amyloid--induced activation of NOD-like receptor family pyrin domain containing 3 (NLRP3) inflammasome (Heneka et al., 2013), and involves microglial activation.  Sentence Unclear

Typos

Line(s)

Typo

notes

25

sys-tems

50

sol-uble

53

organ-isms

59

re-cycling

61

capaci-ty

74

ag-ing

76

ef-fects

105

patho-logical

107

inter-actions

112

nerv-ous

120

ul-timately

126

com-pounds

130

coun-teract

134

per-formance

148

a

and(?)

142

pal-mitoleic

167

ripen-ing

168

-glycosidases

β

188

be-come

203

ef-fect

207

gastrointesti-nal

232

 hy-drolytic

234

glyco-sylated

244

facilitat-ed

251

en-terocyte

259

or-gans

267

vari-ous

269

responsi-ble

305

adminis-tered

315

3H

3H

323

af-ter

356

iden-tify

358

admin-istration

389

pro-duced

391

Endog-enous

393

en-zyme

398

re-ductase

407

Hash-imoto et al.

411

evaluat-ed

420

poten-tial

435

amyloid- -induced

439

-synuclein

 α?

442

(TNF- ?

α

443

fac-tors

446

(H2O2)

(H2O2)

452

de-creases

454

neuroinflam-mation

458

 downregulat-ing

464

im-proved

482, 491,494,499

-amyloid (A peptide)

482

re-spectively

493

pro-tein

494

-secretase

α?

501

A deposition

Amyloid

504

evi-dence

509

pancreatic- cells

β

511

deposi-tion

511,513

-sinuclein

α?

515

evi-dence

518

Nardi-ello

531

pre-venting

533

cyto-plasmic

535

accumula-tion

550

pre-cede

552

in-tracytoplasmic

554

nucleus basal-is

594

endoge-nouse

603

po-tent

630

Pro-tein

663

con-sisted

666

TNF-?

692

en-zymes

700

of hunting-tin

718

subgranu-lar

740

treat-ment

763

occlu-sion

799

pro-duction

807

per-formed

813

antidepres-sants

816

hyperac-tive

821

 TNF-?  IL1 and IFN-?

 α ? β? ϒ? which one?

827

dis-plays

846

degenera-tion

878

neuropro-tection

882

peroxida-tion

905

Peng et al., 2015)

 (Peng et al., 2015)

911

overex-press-ing

913

100   M HTyr

 mM (?) μM (?), nM (?)

917

to water

to H2O

918

(O-2)

O2-

930

(H2O2) in H2O

(H2O2) in H2O

933

(HO2   O-2

 (HO2     O2-

933

water solution

aqueous  solution

943

produc-tion.

955

inacti-vates

961

re-covery

980,982

olecantal

olecanthal

991

indicat-ed

1000, 1028

TNF- IL1

α ?   β?......?

1012

is-chemia

1017

neuroprotec-tive

1033

(H2O2

(H2O2

1035

ni-trohydroxytyrosyl

1055

possi-bility

1054

moie-ties

1065

upregula-tion

1069

secre-tory

1078

cul-tures

1093

can-cer

1098

Varela-Eirín et al.

Varela-Eirín et al. (2020)

1101

senes-cence

1126

in-creasing

1169

re-dox

1179

mu-scle

1232

produc-tion

1382

iso-lated

1391

Giacometti  and colleagues.

Giacometti et al. (2020)

1441,1444,1464,1477

mi-crobiota

1468

re-ported

1480

pro-portion

1507

be-tween

1508

cen-tral

1511

regula-tion

1513

immun-ity

1517

ef-fect

1558

 inflam-matory

1560

neuropro-tection

1565

micro-biota

1567

un-changed

1618

in-hibiting

1624

po-tential

1626

treat-ment

1631

patholo-gies.

1634

metabolic disorders

Include age-related  altered bone metaboilsm

1637

inflamma-tion

1646

evi-dence

1650

appli-cation

The review article by et al. Micheli L. et al.” Role of hydroxytyrosol and oleuropein in the prevention of aging and related disorders: focus on neurodegeneration, skeletal and muscle dysfunction and gut microbiota” summarizes the results of studies that investigate the effects exerted by EVOO polyphenols, in particular hydroxytyrosol (HTyr) and oleuropein (Ole), on several age-associated neurodegenerative diseases. Although the topic appears timely and may  be of interest  to the readers of Nutrients, the revision process  has arisen several critical issues that need to be addressed by the authors in order to make this review article more focused and to increase its impact on the related field.

General Comments

Introduction

Lines102-123 Among aging-related diseases, Authors should also consider some pathological conditions associated with altered bone remodeling processes including osteoarthritis, osteoporosis and other disorders (see for instance Edwards et alBone 201580, 126–130; Guo et al. Sig. Transduct Target Ther 7,391 (2022); Chin et al. J Clin Med. 2022;11(21):6434). In particular, idiopathic osteoporosis of the elderly, which is a significant  risk factor for fragility fractures, is a  relevant global public health problem, being the economic burden of osteoporosis-related fracture enormous. (Clynes  et al,. Br  Bull. 2020;133(1):105-117). Although the current therapeutic options based on the use of anti-resorptive and/or anabolic agents, are effective in preventing bone loss, there is increasing concern about their long-term safety (see Skjødt  et al. e. Br J Clin Pharmacol. 2019 Jun;85(6):106). Hence, the need to find  new molecules endowed with  low  toxicity and limited side effects.  In this setting, growing experimental studies have shown that HTyr and OLE can target several signalling pathways involved in the modulation of bone remodelling processes in normal and pathological conditions including malignant bone diseases (see for instance  Casado-Díaz et al., Food Funct. 8 (2017) 1254; Castejón et al. , Antioxidants (Basel). 9 (2)  (2020) 149; Nicolin V et al. Front. Endocrinol. (Lausanne) 10 (2019) 494); Scoditti et al. Arch. Biochem. Biophys. 527 (2012) 81; Torre E. , Phytochem. Rev. 16 (2017), 1183–1226; M. Svobodova et al , Genes Nutr. 9 (2014) 376;  Xiong ZC Drug Des. Develop. and Ther. 13 (2019) 1879.). These findings suggest  a possible therapeutic role of these molecule in the prevention and  treatment  of various  bone disorders. In line with these observations recent preclinical and clinical studies have provided evidence on the beneficial effects exerted by HTyr and OLE in the prevention and treatment of  age-related bone disorders (see for instance Hagiwara et al. Eur. J. Pharmacol. 662 (2011) 78–84; Chin KY et al. , Int. J. Environ. Res. Public Health 13 (2016); Chin KY et al.,  Nutrients 9 (2017);  Nicolin et al. Front. Endocrinol. (Lausanne) 10 (2019) 494),   including cancer-related bone diseases (J.M. Morana, Nat. Prod. Commun. 11 (2016) 491–492; Leto et al.  Life Sci  2021, 1;264:118694; Przychodzen et al, Anticancer Res. 2019 39(3):1243-1251).

Authors should revise this section according to the new concept    emerged  following the revion of the paper.

2. Chemistry of hydroxytyrosol (HTyr) and oleuropein (OLE)

Lines 129- 130  several types of cancer, inflammation, cardiovascular and neurodegenerative diseases insert some reference related to these studies)

line 169  producing glucose,  Ole aglycone, HTyr and elenolic acid (Scheme 1)

3. Pharmacokinetics of HTyr and OLE

 3.1  Lines 200-229 .  The general aspects of  ADME should be omitted. It is presumed that these concepts are well known by the readers.

Line 236: resident time (?) : residence time

Line 245 and the arrives: better reaches

Line 268 Its capillary  (?) distribution: it would mean widespread distribution?

3.3. Metabolism and distribution

There are some important  aspects of Ole metabolism that need to be to highighted and   in particular :

1)   The potential therapeutic effectiveness of  this molecule is closely related to the possibility that it may reach in adequate concentrations its specific molecular targets in human tissues.  The probability of reaching its  key molecular targets in human tissues at a sufficient dose, is related to its metabolism and bioavailability. In fact, Ole and HTyr undergo extensive phase I and phase II metabolic processes that may affect their bioavailability and the systemic transfer at adequate concentrations to the target tissues, ultimately blunting their therapeutic effectiveness.  In order to overcome these hurdles, an increasing number of studies are currently  undertaken with the aim to develop new semi-synthetic derivatives endowed with abetter bioavailability, improved biological activity and, novel drug delivery systems based on  nanotechnology  (see Bonechi et al  Biophys. Chem. 246 (2019) 25D;  De Luca. Et al.Polymers (Basel). 2022 Apr 23;14(9):1726.  Karković Markovićet al. Molecules. 2019;24(10):2001.Monteiro  et al. .Antioxidants (Basel). 2021;10(3):444 Palagati et al. Daru J. Pharm. Sci. 27 (2019) 695 -708,  Russo et al. J. Pharm. Pharmaceutics 3(1) (2016) 40)

2)   Malliou et al. have recently shown that the a long-term  oral administration of Ole to  male 129/Sv WT mice  induced the synthesis of  the major cytochrome P450s (P450s)  in the liver via activation of PPAR α and other cellular factors, such as AHR, CAR, RXR, and PXR (Malliou et al. Drug Metab Dispos. 2021;49(9):833). This effect could modify the pharmacokinetic profile of co-administered drug whic are substrates of the P450s, thus altering their therapeutic efficacy and toxicity.  Therefore, these effects  shoud be taken into account and in particular  when Ole is co- administered  with drugs with  narrow therapeutic index

3.5. Toxicity

Authors  should discuss the implications of the metabolic aspects of  Ole and Htyr reported above in relation  to a possible future clinical use of these molecules in  differentage-related diseases. Beside, the possibility to take advantage of the various effects of Htyr and Ole on normal and malignant human cells  for therapeutic purposes should be also briefly mentionrd

Section 4 to Section 10

The effects exerted by  Ole  in some of  the  aging-associated  diseases  have not been  mentioned in detail.  An exaustive updated description of  the therapeutic potenial  of Ole in the treatment  of  acute and chronic neurodegenerative and neuropsychiatric disorders as well on mechanisms by which Ole may prevent neurodegeneration are provided by some recent excellent  review  articles ((see Angeloni et al. Int.  J.  Mol.  Sci, 18(11), 2230,2017  Butt et al.  J Food Biochem. 2021;45:e13967. See also Achour et al. Int. J. Mol. Sci. 2016;17:129; (Singh et al Sci Rep. 2023 Feb 11;13(1):2452); Giacometti and Grubić-Kezele Oxid Med Cell Longev. 2020 Jul 30;2020:6125638. Authors should integrate and discuss these  informations.

13. Effect of HTyr and OLE in gut microbiota-brain axis

Emerging experimental and clinical evidence indicates that gut dysbiosis  may have a role in the pathogenesis Ischemic Stroke ( see Chidambaram Cells. 2022 Apr 6;11(7):1239) Wang et al. Front Immunol. 2022 Mar 28;13:845243) as a restoration of the gut microbiome usually improves stroke treatment outcomes by regulating metabolic, immune, and inflammatory responses via the gut–brain axis (GBA). Ole has  been reported to be endowed with  antimicrobial activity against a wide number of  bacteria, (either gram+ or gram-) mycoplasma, and viruses  (Omar, S.H.. Sci. Pharm. 201078;133 Farràs M,. Nutrients. 2020 Jul 23;12(8):220). These findings imply that  Ole and Htyr  may be of potential clinical interest  for  the treatment of various  age-related disorders in which gut  dysbiosis may have a role including neurodegenerative diseases, cardiovascular diseases, metabolic diseases, musculoskeletal diseases, and immune system diseases and cancer (  see Chidambaram 2022; Varesi A, Int. J. Mol. Sci. 2022; 23(20):12289. Di Meo et al., Curr. Drug Metab. 19 (6), 2018;  Memmola. Nutrients. 2022 Sep 10;14(18):3749. )

14. Conclusions

Authors should revise this section according with the new  issue emerged  following the revion of the paper 

References

López-Otín C, Blasco MA, Partridge L, Serrano M, Kroemer G. The hallmarks of aging. Cell. 2013 Jun 6;153(6):1194-217. doi:

10.1016/j.cell.2013.05.039.

 Lines 52-55 and 2196  Ref 192 López-Otín C, Blasco MA, Partridge L, Serrano M, Kroemer G. The hallmarks of aging. Cell. 2013 Jun 6;153(6):1194-217. There is an updated version of this article  (López-Otín  C.  et al. Cell. 2023 Jan 19;186(2):243-278. doi: 10.1016/j.cell.2022).  These authors report that the  number of hallmark of aging should be updated to 12 ( as compared with the previous number, e.g.,9)

Minor point

Lines 435-437….  as also indicated by the amyloid--induced activation of NOD-like receptor family pyrin domain containing 3 (NLRP3) inflammasome (Heneka et al., 2013), and involves microglial activation.  Sentence Unclear

Typos

Line(s)

Typo

notes

25

sys-tems

50

sol-uble

53

organ-isms

59

re-cycling

61

capaci-ty

74

ag-ing

76

ef-fects

105

patho-logical

107

inter-actions

112

nerv-ous

120

ul-timately

126

com-pounds

130

coun-teract

134

per-formance

148

a

and(?)

142

pal-mitoleic

167

ripen-ing

168

-glycosidases

β

188

be-come

203

ef-fect

207

gastrointesti-nal

232

 hy-drolytic

234

glyco-sylated

244

facilitat-ed

251

en-terocyte

259

or-gans

267

vari-ous

269

responsi-ble

305

adminis-tered

315

3H

3H

323

af-ter

356

iden-tify

358

admin-istration

389

pro-duced

391

Endog-enous

393

en-zyme

398

re-ductase

407

Hash-imoto et al.

411

evaluat-ed

420

poten-tial

435

amyloid- -induced

439

-synuclein

 α?

442

(TNF- ?

α

443

fac-tors

446

(H2O2)

(H2O2)

452

de-creases

454

neuroinflam-mation

458

 downregulat-ing

464

im-proved

482, 491,494,499

-amyloid (A peptide)

482

re-spectively

493

pro-tein

494

-secretase

α?

501

A deposition

Amyloid

504

evi-dence

509

pancreatic- cells

β

511

deposi-tion

511,513

-sinuclein

α?

515

evi-dence

518

Nardi-ello

531

pre-venting

533

cyto-plasmic

535

accumula-tion

550

pre-cede

552

in-tracytoplasmic

554

nucleus basal-is

594

endoge-nouse

603

po-tent

630

Pro-tein

663

con-sisted

666

TNF-?

692

en-zymes

700

of hunting-tin

718

subgranu-lar

740

treat-ment

763

occlu-sion

799

pro-duction

807

per-formed

813

antidepres-sants

816

hyperac-tive

821

 TNF-?  IL1 and IFN-?

 α ? β? ϒ? which one?

827

dis-plays

846

degenera-tion

878

neuropro-tection

882

peroxida-tion

905

Peng et al., 2015)

 (Peng et al., 2015)

911

overex-press-ing

913

100   M HTyr

 mM (?) μM (?), nM (?)

917

to water

to H2O

918

(O-2)

O2-

930

(H2O2) in H2O

(H2O2) in H2O

933

(HO2   O-2

 (HO2     O2-

933

water solution

aqueous  solution

943

produc-tion.

955

inacti-vates

961

re-covery

980,982

olecantal

olecanthal

991

indicat-ed

1000, 1028

TNF- IL1

α ?   β?......?

1012

is-chemia

1017

neuroprotec-tive

1033

(H2O2

(H2O2

1035

ni-trohydroxytyrosyl

1055

possi-bility

1054

moie-ties

1065

upregula-tion

1069

secre-tory

1078

cul-tures

1093

can-cer

1098

Varela-Eirín et al.

Varela-Eirín et al. (2020)

1101

senes-cence

1126

in-creasing

1169

re-dox

1179

mu-scle

1232

produc-tion

1382

iso-lated

1391

Giacometti  and colleagues.

Giacometti et al. (2020)

1441,1444,1464,1477

mi-crobiota

1468

re-ported

1480

pro-portion

1507

be-tween

1508

cen-tral

1511

regula-tion

1513

immun-ity

1517

ef-fect

1558

 inflam-matory

1560

neuropro-tection

1565

micro-biota

1567

un-changed

1618

in-hibiting

1624

po-tential

1626

treat-ment

1631

patholo-gies.

1634

metabolic disorders

Include age-related  altered bone metaboilsm

1637

inflamma-tion

1646

evi-dence

1650

appli-cation

Reviewer 2 Report

REPORT FOR AUTHORS: nutrients-2262015

The manuscript ‘Role of hydroxytyrosol and oleuropein in the prevention of ageing and related disorders: focus on neurodegeneration, skeletal muscle dysfunction and gut microbiota’ provides comprehensive information about the general and updated characteristics of hydroxytyrosol and oleuropein, and its effect on neuro-diseases and involvement on gut axis.

The authors have made a huge effort to clarify and synthesize the general characteristic of hydroxytyrosol and oleuropein. In my opinion, the review has a full collection of new references and data from the last few years. Including the tables, which show a very useful compilation of HTyr and OLE in neurodegenerative diseases.

MAJOR REVISION

There are some sections where OLE is not referred to any pathology, e.g. Parkinson’s disease, but there is recent literature that can be included (e.g. doi: 10.3390/ijms21072588). The authors should include more information about OLE in neuro-disease. Can authors include some justification due to the scarcity or lack of literature in this regard? 

In this sense, does OLE exert any neuroprotective function in neurodegenerative diseases? Recently, Butt MS et al 2021, published the neuroprotective effects of oleuropein (doi: 10.1111/jfbc.13967). Also, regarding retinopathies in section 10.4; no information is included about OLE. Recently, Zheng S et al, 2021, claim the efficacy and mechanisms of OLE in mitigating diabetes and diabetes complications, including diabetic retinopathy (doi: 10.1021/acs.jafc.1c01404)

I guess there is an unconscious tendency to describe all HTyr beneficial effects. Despite being a product of the enzymatic hydrolysis of OLE, it is opportune to include OLE results or at least a justification of the scarce literature.

MINOR REVISION

Line 35 to line 39: Add reference

Line 53 and line 61: Organ-ism, and capaci-ty. Check the writing through the paper, many words that are hyphen separated. 

Line 161: Add reference

Line 446: H2O2 change to H2O2. Check through the paper. Also, for other oxidative reactive species (e.g. superoxide anion (Line 933))

Line 477: Section 6. Neuroprotective function of HTyr in neurodegenerative diseases. Should include the word “OLE”

Line 926 and 935: Nox2 and NOX2. Check through the paper the abbreviation for each enzyme/protein referred, since there is no consistency in the spelling.

Round 2

Reviewer 2 Report

REPORT FOR AUTHORS: nutrients-2262015

The authors of the manuscript ‘Role of hydroxytyrosol and oleuropein in the prevention of ageing and related disorders: focus on neurodegeneration, skeletal muscle dysfunction and gut microbiota’ have improved the quality of the review presented and solved the major point addressed in the first peer review, which aimed to the authors to include more information about OLE in neurodegenerative diseases.